# Dynamic optical response of solids following 1-fs-scale photoinjection

Dmitry A. Zimin[1,2], Nicholas Karpowicz[1,3 ✉], Muhammad Qasim[1,2], Matthew Weidman[1,2], Ferenc Krausz[1,2] & Vladislav S. Yakovlev[1,2 ✉]

Photoinjection of charge carriers profoundly changes the properties of a solid. This manipulation enables ultrafast measurements, such as electric-field sampling[1,2], advanced recently to petahertz frequencies[3–7], and the real-time study of many-body physics[8–13]. Nonlinear photoexcitation by a few-cycle laser pulse can be confined to its strongest half-cycle[14–16]. Describing the associated subcycle optical response, vital for attosecond-scale optoelectronics, is elusive when studied with traditional pump-probe metrology as the dynamics distort any probing field on the timescale of the carrier, rather than that of the envelope. Here we apply field-resolved optical metrology to these dynamics and report the direct observation of the evolving optical properties of silicon and silica during the first few femtoseconds following a near-1-fs carrier injection. We observe that the Drude–Lorentz response forms within several femtoseconds—a time interval much shorter than the inverse plasma frequency. This is in contrast to previous measurements in the terahertz domain[8,9] and central to the quest to speed up electron-based signal processing.

The nonlinear interaction of an intense laser pulse with a medium changes its refractive index. These changes may disappear immediately after the end of the pulse, as in the case of the optical Kerr effect. They may also outlast the pulse if electrons are promoted to excited states. Notably, nonlinear processes are confined to a time interval shorter than the pulse duration, multiphoton excitation being a prominent example[17]. Such a nonlinearity allows the shortest light pulses, with a single dominant half-cycle of the electric field, to photoinject most charge carriers within a fraction of that half-cycle. The resultant extreme temporal confinement of carrier injection opens new opportunities for ultrafast science[18–23].

Here we investigate how the optical response of this electron–hole plasma forms after sudden, 1-fs-scale, photoexcitation of valence electrons. The time required for the response to build up depends not only on the duration of the laser pulse that creates the plasma, but also on how long it takes quasiparticles to acquire their properties. Collective behaviour, such as Coulomb screening and plasma scattering, is expected to form on the timescale of the inverse plasma frequency, henceforth called the plasma period[8,24–27]. Thus, a question of far-reaching implications is whether the plasma frequency sets a fundamental speed limit for future advancement of optoelectronic signal processing and metrology. The time it takes the plasma response to emerge is also essential for modelling the interaction of solids with intense laser pulses of a few cycles. This highly nonlinear interaction may populate states in hundreds of energy bands, which presents formidable challenges for an ab initio description of many-electron dynamics. Thus, efficient and accurate modelling of the physical processes underlying cutting-edge metrology and signal processing critically depends on the role that many-body physics plays during the first few femtoseconds after photoexcitation.

## Field-resolved pump-probe measurements

In our experiments, a 3 fs (full-width at intensity half-maximum) linearly polarized near-infrared pump pulse (with a period of carrier frequency of 2.3 fs) photoexcited silicon (with a direct band gap of 3.2 eV) and fused-silica (with an energy gap of 9 eV) by multiphoton absorption. The arrival time, $\tau$, of this pump pulse was varied with respect to a weak 12 fs test pulse carried at a central wavelength of 2.1 μm (with a period of carrier frequency of 7 fs). For Si, photoinjection was confined to approximately 2 fs, as shown by the green curve in Fig. 1b, which represents the nonlinear work performed by the pump field. This time-dependent work is mostly spent on creating charge carriers, and we calculated its value using time-dependent density functional theory[28]. The relatively long period of the test field is beneficial for studying light-driven electron motion. To decouple this motion from the photoinjection, the pump and test pulses were polarized orthogonally. The transmitted pump pulse was blocked using a wire-grid polarizer, while the oscillating electric field (henceforth, waveform) of the test pulse was recorded using the recently developed generalized heterodyne optical sampling technique (GHOST)[29].

Figure 2 displays waveforms transmitted through a 230-nm-thick silicon sample. In Fig. 2a, we compare the reference waveform, $E_{ref}(t)$, which is the transmitted test pulse without photoinjection, with the waveforms $E(t)$ recorded for two arrival times of the 0.8 V Å$^{-1}$ pump pulse: before and at the centre of the test pulse. In the latter case, the waveform remains unchanged until photoinjection. Figure 2b plots the change in the transmitted test field (with and without carrier injection) for different moments of injection. This change, $\Delta E(t) = E(t) - E_{ref}(t)$, builds up in the first half-cycle of the test field (that is, within 3–4 fs) following the peak of the pump pulse and starts decaying another half-cycle later.

[1]Max-Planck-Institut für Quantenoptik (MPQ), Garching, Germany. [2]Ludwig-Maximilians-Universität München (LMU), Garching, Germany. [3]CNR NANOTEC Institute of Nanotechnology, Lecce, Italy. ✉e-mail: nicholas.karpowicz@mpq.mpg.de; vladislav.yakovlev@mpq.mpg.de

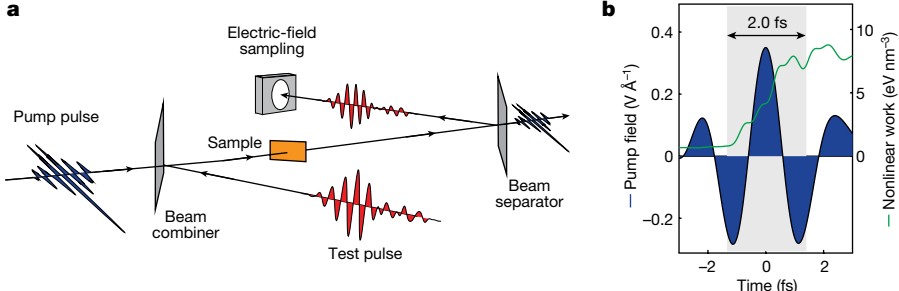

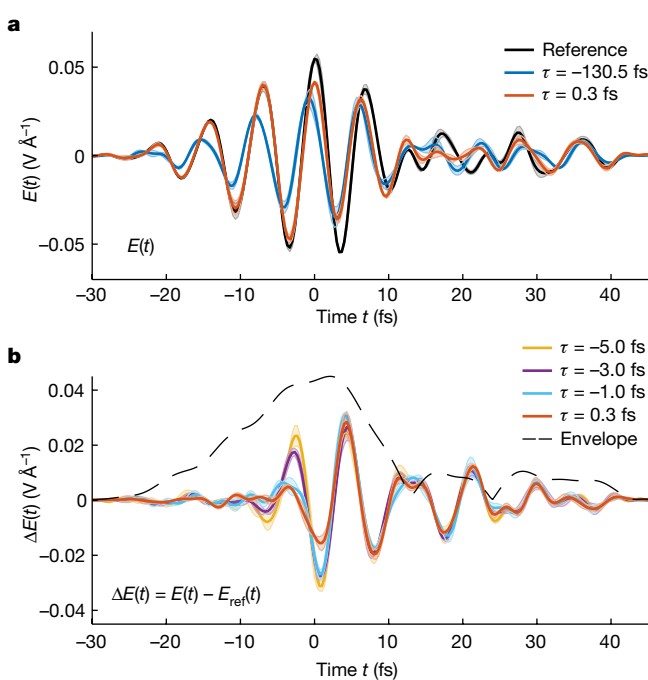

**Fig. 1 | Field-resolved detection of charge-carrier motion controlled by sub-half-cycle photoinjection. a**, A schematic of the pump-probe measurements in which the waveform of the test pulse (red) was measured after transmission through a thin sample (orange) for different delays of the pump pulse (blue), which created charge carriers by multiphoton absorption. **b**, The nonlinear nature of the photoinjection confines its duration to a time interval shorter than the half wave-cycle of the test pulse, even for the weaker

(two-photon) nonlinearity occurring in Si (shown in the panel). The green curve shows the nonlinear work performed by the pump field. The grey area highlights the time between the moments at which the work reaches 10% and 90% of its final value. In $SiO_2$, this confinement is even stronger owing to the higher number of photons needed for promoting a valence electron to the conduction band[16].

The experiment provides a textbook example for how the material polarization (in this case induced by the injected charge carriers) is imprinted in the electric field transmitted through the sample.

The measured photoexcitation-induced waveform distortions carry information about how the optical thickness and absorption of the sample change with time. In the static case, these properties are fully described by a complex refractive index. In the Methods, we define a generalized refractive index that describes the medium's dynamic response. For a dispersive medium, such a generalized refractive index depends on both frequency, $\omega$, and time, $t$. The formation of an electron–hole plasma decreases the real part of the refractive index, which increases the phase velocity of the test pulse and thus shifts its carrier wave to earlier times. The photoinjected electrons and holes also absorb light, which attenuates the test waveform transmitted after the pump pulse. We clearly see both these effects in Fig. 2a. From Fig. 2b we see that already 3–4 fs after the peak of the pump pulse the waveform distortions measured for slightly different delays merge within their standard deviations. We interpret this as a transition from ballistic to non-ballistic conduction.

## Drude–Lorentz model

For gaining insight into how ultrafast photoinjection reshapes the test pulse, it is sensible to seek the simplest possible model able to account for the essential physics. For this purpose, we use the Drude–Lorentz model in the idealized case of instantaneous photoinjection. A textbook derivation of the Drude–Lorentz model[30] considers the classical motion of free charge carriers and bound electrons in an oscillating electric field. In the Methods, we summarize and state again this derivation for the specific case in which pairs of charge carriers are created and begin their classical motion under the influence of the electric force of the test field at a certain moment $\tau$ (similar models were previously developed for terahertz time-domain spectroscopy[31–33]). The result is a complex time-dependent refractive index that describes the properties of the electron–hole plasma but does not account for wave-mixing processes that require the presence of the pump pulse, such as the optical Kerr effect or the dynamical Franz–Keldysh effect[34,35]. On the basis of this time-dependent refractive index, we investigate the extent to which the observed waveform distortion can be accounted for by the Drude–Lorentz model.

To this end, we numerically optimize the phenomenological parameters of the model to fit the measured waveform distortions in the time interval $t \geq \tau + 5$ fs. In this time window, we can neglect any nonlinear interaction of the 3 fs pump pulse with charge carriers that could induce a polarization response in the direction of the test field.

The results are shown in Fig. 3, which compares the measurements with the 230-nm-thick silicon sample to those with a 12.7-μm-thick fused-silica sample, for which the pump pulse had a peak field strength of 1.4 V Å$^{-1}$. In Fig. 3e,f, we plot the difference between the measured and reconstructed waveform distortions. For both materials, the model accurately reconstructs the measured delay-dependent waveform distortion except in a time interval of few femtoseconds around the centre of the pump pulse. In the fused-silica measurements (Fig. 3f), we evaluate an upper limit for the formation time of the Drude–Lorentz response to be 4 fs. This is notable given that the plasma period, which we retrieve from these measurements, is 110 fs. For silicon, we estimate

**Fig. 2 | The effect of ultrafast photoinjection on the test pulse transmitted through the 230-nm-thick silicon sample. a**, The reference waveform $E_{ref}(t)$ (black), which is the electric field of the test pulse transmitted through the unperturbed sample, is compared with a test waveform with photoexcitation preceding the test pulse (blue) and the waveform measured with the 3 fs pump pulse arriving at $t = 0$ (orange). The shaded areas represent the standard deviations from three independent measurements. **b**, The measured photoexcitation-induced change in the transmitted electric field, $\Delta E$, for several arrival times of the pump pulse, $\tau$. The dashed curve shows the envelope of $E_{ref}(t)$.

the plasma period to be between 2 and 3 fs, which is comparable to the duration of carrier injection.

## Time-dependent refractive index

Whereas the interaction with the pump pulse is nonlinear, the polarization response to the weak test pulse is linear. So, its transmission through the sample is fully described by the complex-valued, time-dependent refractive index $n_\omega(t)$, which does not depend on the shape of the test waveform and encodes all the information about the physical processes that make the medium's optical properties evolve. It is possible to retrieve the time-dependent refractive index from the waveform distortions. We find, however, that a good first approximation to the time-dependent refractive index at the central frequency of the test pulse may be obtained without sophisticated reconstruction by applying the following analysis to delay-dependent waveforms.

A mere change in the optical thickness of a sample barely affects half-cycle amplitudes but it shifts the zero crossings of the electric field in time. At the same time, changes in absorption and reflection strongly affect the half-cycle amplitudes, but not the positions of the field's zero crossings. This indicates that the delay dependence of zero-crossing shifts should be closely related to $\mathrm{Re}[n_\omega(t)]$, whereas the delay dependence of the half-cycle amplitudes should be mainly determined by $\mathrm{Im}[n_\omega(t)]$. According to our modelling, this indeed is the case in our measurements. The error bars in Fig. 4 show how photoinjection shifts the zero crossings and reduces the half-cycle amplitudes in the measured data. These error bars represent standard deviations evaluated from delay scans that were performed one after another (three scans for silicon and five scans for fused silica).

The solid curves in Fig. 4 were obtained by applying the same analysis to the reconstructed waveforms, for which we used the Drude–Lorentz model with the same parameters as before (in Fig. 3). The real and imaginary parts of the refractive index in this model are depicted with empty circles and squares, respectively. More accurately, this time-dependent refractive index describes the propagation of a monochromatic test wave with a wavelength of 2.1 μm, which is the central wavelength of the test pulse. We see that the shape of $\mathrm{Re}[n_\omega(t)]$ closely resembles that of the zero-crossing shift, whereas the shape of $\mathrm{Im}[n_\omega(t)]$ matches that of the half-cycle amplitude reduction. We also observe here that neither the theoretical refractive index nor the reconstructed waveforms change abruptly even though photoinjection was modelled as an instantaneous event. This is because, in such measurements, $\Delta E(t)$ is approximately proportional to the electric current induced by the test field[36], and the current gradually builds up as the field accelerates the charge carriers. For the same reason, the time dependence of $n_\omega(t)$ does not end with photoinjection (see equation (16) in the Methods).

With these insights, Fig. 4 allows us to further analyse the part of the medium response that is not accounted for by our Drude–Lorentz model with instantaneous photoinjection. In the case of silicon (Fig. 4a), the modelled change in the refractive index is steeper than that inferred from the measurements: measured between 10% and 90% of the maximum change, the rise time of absorption is 5.2 fs in the measurements and 1.4 fs in the simulations. This discrepancy can be largely explained by the approximation of instantaneous photoinjection that we made in our model. Even if we neglect phonon-assisted single-photon transitions, we expect that 80% of the charge carriers are photoinjected within a time interval of approximately 2 fs (Fig. 1b). We conclude that the results in Fig. 4a do not contain any unambiguous evidence for many-body dynamics prolonging the buildup of the Drude–Lorentz response.

In the case of fused silica (Fig. 4b), photoinjection requires the absorption of at least four photons from the pump pulse. Hence, we expect that more than 70% of the charge carriers appeared during the central half-cycle of the pulse. Here the approximation of instantaneous photoinjection is more applicable, and other effects are responsible

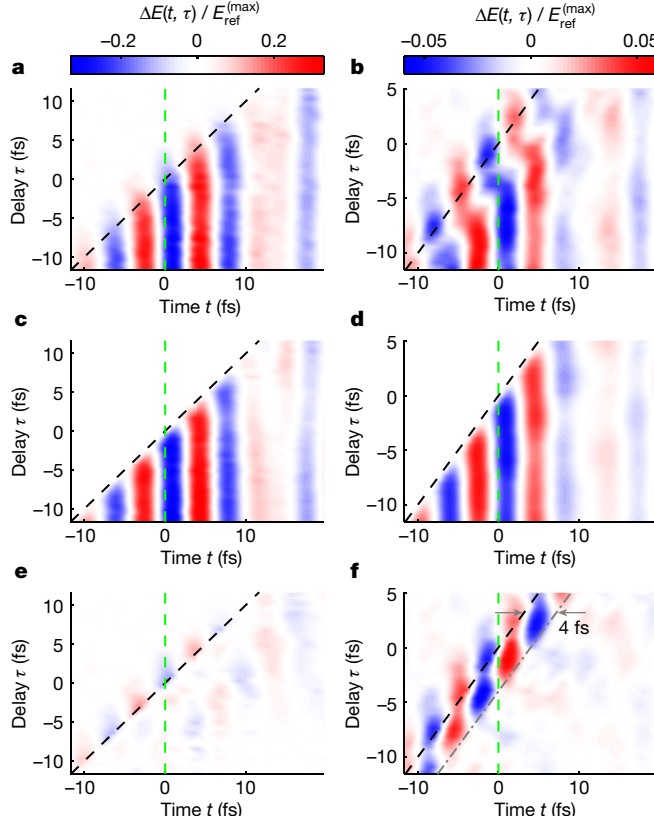

**Fig. 3 | Distortion of the test waveforms induced by the pump pulse.**
These pseudocolor diagrams show $\Delta E(t, \tau)$ normalized to the peak value of the reference field, $E_{\mathrm{ref}}^{(\mathrm{max})}$. The black dashed lines show the middle of the pump pulse ($t = \tau$). The green dashed lines represent $t = 0$, at which the reference waveform reaches its maximal value. **a,b**, $\Delta E(t, \tau)$ measured for silicon (**a**) and silica (**b**). **c,d**, $\Delta E(t, \tau)$ reconstructed using the Drude–Lorentz model for silicon (**c**) and silica (**d**). **e,f**, By subtracting the reconstructed $\Delta E(t, \tau)$ from the measured one, we obtain the non-Drude–Lorentz waveform distortions, which are shown for silicon (**e**) and silica (**f**), using the same colour schemes as those in the upper panels.

for the differences between the error bars and the solid curves. The test field's zero crossings first get delayed and only then experience a negative time shift. We associate this transient increase in the optical thickness of the sample with the optical Kerr effect. We cannot fully explain why the positive zero-crossing shifts extend over 8 fs, which is three times as large as the full-width half-maximum of the pump pulse, but the propagation through the 12.7 μm sample may be partially responsible for it. We had to use a relatively thick fused-silica sample to achieve an acceptable signal-to-noise ratio in spite of the miniscule changes in the refractive index—the relative inefficiency of the high-order multiphoton absorption in fused silica puts an upper limit on the concentration of charge carriers that can be achieved without destroying the sample. In contrast to silicon, the contribution from the electron–hole plasma to $\Delta n$ in fused silica was comparable in magnitude to that from wave-mixing processes enabled by the presence of the pump pulse. In addition to the optical Kerr effect, these processes include a transient increase in absorption, which is a manifestation of the Franz–Keldysh effect (the Keldysh parameter was close to 1 in the fused-silica measurements).

## Conclusions

In summary, petahertz-scale optical-field metrology in a pump-probe setting enables the direct observation of how the optical properties

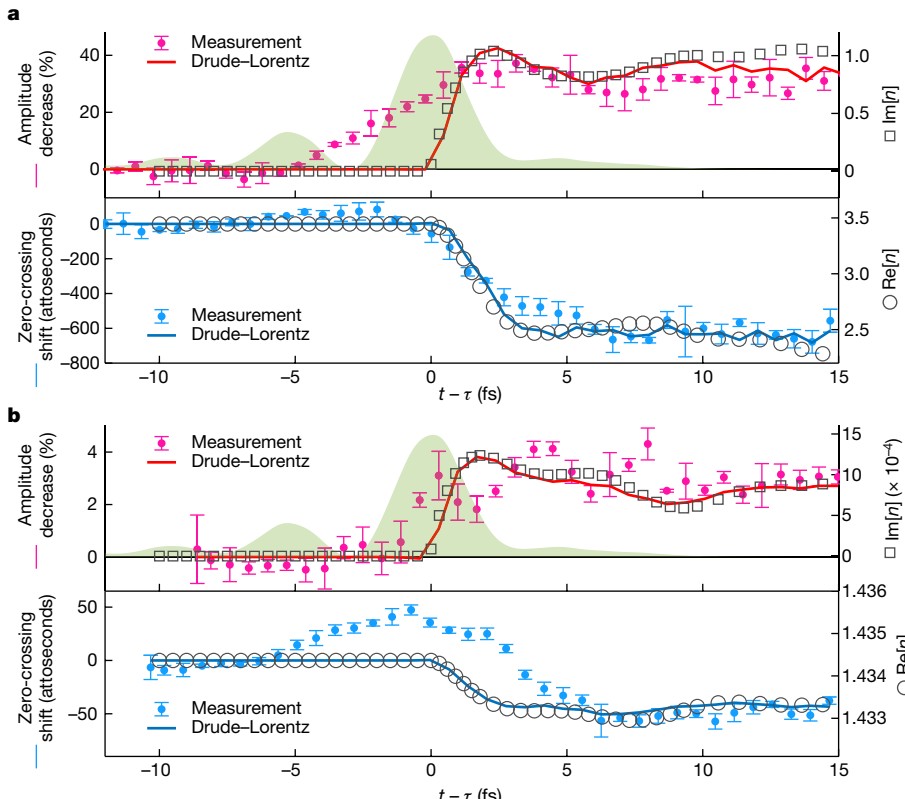

**Fig. 4 | Time dependence of optical properties. a**, The analysis of the silicon data. The positions of the magenta error bars and the red curve show the photoinjection-induced decrease of the half-cycle amplitudes in the measured and reconstructed waveforms, respectively. These data are juxtaposed with the imaginary part of the refractive index (black squares), evaluated using the Drude–Lorentz model. The positions of the light-blue error bars and the blue curve represent the change in the timing of the test field's zero crossings in the measured and reconstructed waveforms, respectively. The black circles show the real part of the time-dependent refractive index. **b**, The same analysis applied to the fused-silica data. The horizontal axis in these plots is the time passed after the middle of the pump pulse. The intensity envelope of the pulse is shown by the green area.

of a medium evolve after 1-fs-scale photoinjection. For a sufficiently thin sample, the time dependence of the real part of the refractive index is closely matched by the delay dependence of the test waveform's zero crossings, whereas the imaginary part of the refractive index matches the delay dependence of the half-cycle amplitudes.

Both in silicon and in fused silica, we observed that the Drude–Lorentz response forms within a few femtoseconds after photoinjection. In fused silica, this time was much shorter than the inverse plasma frequency. Under the premise that the plasma frequency sets the relevant timescale for the formation of charge–charge interactions in many-body systems, we may conclude that many-body phenomena have little effect on how a photoexcited wide-gap material, in this case fused silica, responds to visible or infrared light. This conclusion grants credibility to treating the interaction of abruptly injected carriers with optical fields in theoretical frameworks in which electrons are described with single-particle mathematical objects[37,38], for example by means of time-dependent density functional theory[39] or semiconductor Bloch equations[40]. At the same time, it is likely that future research will find examples in which collective processes are pronounced in suddenly photoexcited solids and petahertz-scale optical-field metrology will be a powerful tool for investigating them.

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

# Methods

## The linear polarization response of a medium with time-dependent properties

When a pulse of light is transmitted through a material, the textbook description of the interaction between the electric field of the pulse and matter is generally given in terms of material parameters, such as the refractive index, dielectric function or susceptibility. When the material changes its properties slowly, the generalization of these quantities is straightforward. A more careful generalization is necessary when the properties substantially change within a single oscillation of the optical field. This is the purpose of this and the following sections.

Let us consider the polarization response induced by a light pulse that is so weak that the relationship between its electric field, $E(t)$, and the induced polarization, $P(t)$, can be assumed to be linear. By definition, a linear response must satisfy two requirements. First, if $E(t)$ induces $P(t)$, then $aE(t)$ must induce $aP(t)$, where $a$ is a constant. Second, if $E_1(t)$ induces $P_1(t)$ and $E_2(t)$ induces $P_2(t)$, then the sum of the fields, $E_1(t) + E_2(t)$, must induce the polarization that is equal to $P_1(t) + P_2(t)$. For simplicity, we regard the electric field and the induced polarization as scalar quantities and we assume that the polarization response is local. When the medium is static, the most general expression of the linear polarization response that satisfies causality is given by the well-known equation

$$P(t) = \int_0^\infty d\tau\, \chi^{(1)}(\tau) E(t-\tau),$$

which we have written in CGS units.

Here $\chi^{(1)}(\tau)$ is the first-order, time-domain susceptibility. As the right-hand side of the above expression is a convolution, the linear polarization response of a static medium looks particularly simple in the frequency domain: $\widetilde{P}(\omega) = \chi^{(1)}_\omega \widetilde{E}(\omega)$.

Let us now generalize this formalism to the case in which the optical properties of the medium evolve in time. The time-domain description is a convenient starting point for such a generalization.

The following equation satisfies the causality principle and the linearity requirements:

$$P(t) = \int_0^\infty d\tau\, \chi^{(1)}(\tau, t) E(t-\tau). \tag{1}$$

The linear susceptibility now depends on two arguments. The first one, $\tau$, is required because the polarization response does not build up instantaneously. The second argument, $t$, represents the time dependence of the medium's properties. Because of this second argument, the Fourier transform of the integral is no longer a product of $E(t)$ and a frequency-domain susceptibility; one must also perform a convolution when working in the frequency domain. This means that a medium with time-dependent properties can mix frequency components of $E(t)$ and generate new ones. At this point, it is instructive to consider two special cases: the impulse response and the response to a monochromatic wave. These cases are useful because any pulse can be written as a linear superposition of delta spikes,

$$E(t) = \int_{-\infty}^\infty dt_0\, E(t_0)\delta(t-t_0) \tag{2}$$

or monochromatic waves,

$$E(t) = \frac{1}{2\pi}\int_{-\infty}^\infty d\omega\, \widetilde{E}(\omega) e^{-i\omega t}, \tag{3}$$

the latter equation being just the inverse Fourier transform. By substituting $E(t)$ with $\delta(t-t_0)$ in equation (1), we get the polarization response to a delta spike that arrives at time $t_0$:

$$P_\delta(t, t_0) = \int_0^\infty d\tau\, \chi^{(1)}(\tau, t)\delta(t-\tau-t_0) = \chi^{(1)}(t-t_0, t)\Theta(t-t_0). \tag{4}$$

Here $\Theta$ is the Heaviside function, which can be incorporated into the definition of $\chi^{(1)}(\tau, t)$. We can rewrite this equation in the form that gives the recipe for evaluating $\chi^{(1)}(\tau, t)$ from the polarization response induced by a delta spike:

$$\chi^{(1)}(\tau, t) = P_\delta(t, t-\tau). \tag{5}$$

We see that $\chi^{(1)}(\tau, t)$ is simply the polarization that the delta spike arriving at time $t - \tau$ induces at time $t$.

Let us now return to equation (1) and substitute $E(t)$ with $e^{-i\omega t}$. This yields the response to a monochromatic wave:

$$P_\omega(t) = \int_0^\infty d\tau\, \chi^{(1)}(\tau, t) e^{-i\omega(t-\tau)} = e^{-i\omega t}\chi_\omega(t), \tag{6}$$

where $\chi_\omega(t)$ is the Fourier transform of $\chi^{(1)}(\tau, t)$ with respect to the first argument:

$$\chi_\omega(t) = \int_0^\infty d\tau\, \chi^{(1)}(\tau, t) e^{i\omega\tau}. \tag{7}$$

We can regard $\chi_\omega(t)$ as a time-dependent linear susceptibility to a monochromatic wave.

Equipped with $\chi_\omega(t)$, we can evaluate the linear polarization response to an arbitrary pulse as

$$P(t) = \frac{1}{2\pi}\int_{-\infty}^\infty d\omega\, \widetilde{E}(\omega) e^{-i\omega t}\chi_\omega(t) \tag{8}$$

In the frequency domain, this translates into

$$\widetilde{P}(\omega) = \frac{1}{2\pi}\int_{-\infty}^\infty d\omega'\, \widetilde{E}(\omega')\widetilde{\chi}_{\omega'}(\omega - \omega'),$$

where

$$\widetilde{\chi}_{\omega'}(\omega) = \int_\infty^{-\infty} dt\, \chi_{\omega'}(t) e^{i\omega t} = \iint_{-\infty}^\infty dt\, d\tau\, \chi^{(1)}(\tau, t) e^{i(\omega t + \omega'\tau)}$$

is the two-dimensional Fourier transform of $\chi^{(1)}(\tau, t)$ with respect to both arguments.

## Time-dependent refractive index

For a static medium, the complex-valued refractive index, $n_\omega$, describes the propagation of a monochromatic wave according to the following formula:

$$E_\omega(z, t) = \mathrm{Re}\left[\exp\left(i\frac{\omega}{c}n_\omega z\right) e^{-i\omega t}E_\omega(0, t)\right],$$

which is consistent with Maxwell's equations as long as $c$ is the vacuum speed of light and $n_\omega^2 = 1 + 4\pi\chi^{(1)}_\omega$ (for a non-magnetic medium). Although it may look reasonable to define a time-dependent refractive index as $\sqrt{1 + 4\pi\chi^{(1)}_\omega(t)}$, this will not necessarily preserve the above expression for pulse propagation. Let us search for a definition of $n_\omega(t)$ that allows us to associate its real and imaginary parts with the optical thickness and the absorption of a thin film. The complex-valued transmittivity, $T_\omega$, of a thin sample relates the incident and transmitted fields: if the incident field is a monochromatic wave $E_\omega^{(0)}e^{-i\omega t}$, then the transmitted field is $T_\omega E_\omega^{(0)}e^{-i\omega t}$. Here the generalization to media with time-dependent optical properties is unambiguous:

$$E_\omega^{\mathrm{out}}(t) = T_\omega(t) E_\omega^{\mathrm{in}}(t) = T_\omega(t) E_\omega^{(0)}e^{-i\omega t}. \tag{9}$$

If $T_\omega$ depends on time, $E_\omega^{out}(t)$ will no longer be a monochromatic wave—the $\omega$ subscript merely indicates the frequency of the incident wave. As long as the transmission is linear, knowing $T_\omega(t)$ allows one to evaluate the transmission of an arbitrary light pulse by decomposing it into monochromatic waves, transmitting each of them separately and assembling the outcomes:

$$E^{out}(t) = \frac{1}{2\pi}\int_{-\infty}^{\infty} d\omega\, T_\omega(t) e^{-i\omega t}\tilde{E}^{in}(\omega)$$
$$= \frac{1}{2\pi}\int_{-\infty}^{\infty} d\omega\, T_\omega(t) e^{-i\omega t}\int_{-\infty}^{\infty} dt'\, e^{i\omega t'} E^{in}(t'). \tag{10}$$

The transmission of a plane-parallel plane with static properties is known to be

$$T_\omega = \frac{1}{\cos(\delta_\omega) - \frac{i}{2}\left(n_\omega + \frac{1}{n_\omega}\right)\sin(\delta_\omega)}, \tag{11}$$

where

$$\delta_\omega = \frac{\omega}{c} n_\omega d,$$

and $d$ is the thickness.

For a generalized time-dependent refractive index to describe the transmission through a thin sample, we define $n_\omega(t)$ via the time-dependent transmission:

$$T_\omega(t) = \frac{1}{\cos(\delta_\omega(t)) - \frac{i}{2}\left(n_\omega(t) + \frac{1}{n_\omega(t)}\right)\sin(\delta_\omega(t))}, \tag{12}$$

$$\delta_\omega(t) = \frac{\omega}{c} n_\omega(t) d. \tag{13}$$

According to this definition, the effective refractive index may depend on the sample thickness, but this dependence disappears in the limit of an infinitesimally thin sample. To show this, we first consider $|\delta_\omega(t)| \ll 1$ and simplify the above two equations:

$$T_\omega(t) = 1 + \frac{i\omega d}{2c}[1 + n_\omega^2(t)]. \tag{14}$$

Let us now relate $n_\omega(t)$ to $\chi_\omega^{(1)}(t)$, which we defined earlier. Light transmitted through a sample is the superposition of the incident wave and that emitted by the electric current induced in the medium. If light propagates along the $z$ axis, while the electric current $J(z,t)$ is confined to $0 \le z \le d$ and flows in a direction perpendicular to the $z$ axis, then Maxwell's equations give the following expression for the electric field induced by the electric current:

$$E^{sheet}(z,t) = -\frac{2\pi}{c}\int_0^d dz'\, J\left(z', t - \frac{|z-z'|}{c}\right).$$

The field of a transmitted monochromatic wave is then given by

$$E_\omega^{out}(t) = e^{i\frac{\omega}{c}d - i\omega t} E_\omega^{(0)} - \frac{2\pi}{c}\int_0^d dz'\, J_\omega\left(z', t - \frac{|d-z'|}{c}\right).$$

For an infinitesimally thin sample, the dependence of $J$ on $z$ can be neglected, and we arrive at

$$E_\omega^{out}(t) = e^{-i\omega t}\left(1 + i\frac{\omega}{c}d\right)E_\omega^{(0)} - \frac{2\pi d}{c} J_\omega(t). \tag{15}$$

By comparing this expression with equation (9), we derive

$$T_\omega(t) = 1 + \frac{i\omega d}{c}\left(1 + \frac{2\pi i}{\omega}\frac{J_\omega(t)}{E_\omega^{(0)}} e^{i\omega t}\right).$$

By comparing this expression with equation (14), we relate the time-dependent refractive index to the electric current density:

$$n_\omega^2(t) = 1 + \frac{4\pi i}{\omega}\frac{J_\omega(t)}{E_\omega^{(0)}} e^{i\omega t}. \tag{16}$$

We see that $n_\omega(t)$ indeed does not depend on $d$.

The electric current density $J_\omega(t)$ is the time derivative of the polarization induced by a monochromatic wave:

$$J_\omega(t) = P'_\omega(t) = \frac{d}{dt}(e^{-i\omega t}\chi_\omega(t)E_\omega^{(0)}) = e^{-i\omega t}E_\omega^{(0)}(\chi'_\omega(t) - i\omega\chi_\omega(t)).$$

Substituting this equation into equation (16), we arrive at the sought after relationship between the generalized refractive index and the generalized linear susceptibility:

$$n_\omega^2(t) = 1 + 4\pi\left(1 + \frac{i}{\omega}\frac{d}{dt}\right)\chi_\omega(t). \tag{17}$$

Although we derived this equation for an infinitesimally thin film, one can still use it in equation (12), as a first approximation, for the transmission of a sample with a small but finite thickness.

## Drude–Lorentz model for instantaneous photoinjection
The optical properties of conducting solids are well approximated by the Drude–Lorentz model. This is a combination of the Drude model, which describes the motion of free charge carriers, and the Lorentz oscillator model, which describes various resonances that shape the polarization response. The rigorous derivation of the Drude–Lorentz model requires a quantum-mechanical treatment, but essentially the same result can be obtained classically. Here we repeat the the classical derivation for the case in which a solid suddenly changes its properties. At this moment, free charge carriers appear, which are accelerated by the electric field of the test pulse according to Newton's second law:

$$x''(t) + \gamma_D x'(t) = -\frac{e}{m_D}E(t). \tag{18}$$

Here $x$ is the electron displacement caused by the external electric field $E(t)$, subscript 'D' stands for 'Drude', $\gamma_D$ is the rate of momentum relaxation, $e > 0$ is the elementary charge and $m_D$ is the average effective mass of charge carriers.

In addition to creating charge carriers, photoexcitation enables interband transitions that could not take place in the unperturbed solid: once an electron makes a transition from a valence state to a state in a conduction band, the electron can then be further excited into a higher conduction band by absorbing a photon, whereas the vacancy left in the valence band can be filled by photoexciting an electron from a deeper valence band. Each such transition gives rise to a separate Lorentz term, but it is often sufficient to consider just a few of them to approximate the linear polarization response in a limited spectral range. In the classical description, the Lorentz contribution to the polarization response emerges from the solution of the following form of Newton's equation for the electron displacement:

$$x''(t) + 2\gamma_L x'(t) + \omega_r^2 x(t) = -\frac{e}{m_L}E(t), \tag{19}$$

where $\omega_r$ is the frequency of a Lorentz resonance, $\gamma_L$ is its relaxation rate, and $m_L$ is an effective mass. For simplicity, we will consider just

one Lorentz term, and we will solve the differential equations with the following initial conditions: $x(t_0) = 0$, $x'(t_0) = 0$, where $t_0$ is the moment of sudden photoexcitation. The polarization response is related to the classical electron displacement via $P(t) = -eNx(t)$. For the Drude response, $N$ is the concentration of charge carriers; for the Lorentz response, it is the concentration of Lorentz oscillators. To write the final result without explicitly using the concentrations and effective masses, we introduce plasma frequencies. In CGS units, their standard definitions are

$$\omega_{\mathrm{pl,D}} = \sqrt{\frac{4\pi e^2 N_{\mathrm{D}}}{m_{\mathrm{D}}}},$$

$$\omega_{\mathrm{pl,L}} = \sqrt{\frac{4\pi e^2 N_{\mathrm{L}}}{m_{\mathrm{L}}}}.$$

We express the polarization response to a monochromatic wave, $e^{-i\omega t}$, as $P_\omega(t) = \chi_\omega(t)e^{-i\omega t}$, solve the above differential equations and obtain the following equations for the time-dependent linear susceptibility in the presence of instantaneous photoinjection:

$$\chi_\omega(t) = \chi_\omega^{(0)} + \Delta\chi_\omega^{\mathrm{Drude}}(t) + \Delta\chi_\omega^{\mathrm{Lorentz}}(t), \tag{20}$$

$$\Delta\chi_\omega^{\mathrm{Drude}}(t+t_0) = -\Theta(t)\frac{\omega_{\mathrm{pl,D}}^2}{4\pi}\left[\frac{1}{\omega^2 + i\gamma_{\mathrm{D}}\omega} + i\frac{e^{i\omega t}}{\gamma_{\mathrm{D}}}\left(\frac{1}{\omega} - \frac{e^{-\gamma_{\mathrm{D}}t}}{\omega + i\gamma_{\mathrm{D}}}\right)\right], \tag{21}$$

and

$$\Delta\chi_\omega^{\mathrm{Lorentz}}(t+t_0)$$
$$= \Theta(t)\frac{\omega_{\mathrm{pl,L}}^2}{4\pi}\left[\frac{1}{\omega_{\mathrm{r}}^2 - 2i\gamma_{\mathrm{L}}\omega - \omega^2} - \frac{e^{-\gamma_{\mathrm{L}}t}e^{i\omega t}}{2\Omega}\left(\frac{e^{-i\Omega t}}{\Omega - \omega - i\gamma_{\mathrm{L}}} + \frac{e^{i\Omega t}}{\Omega + \omega + i\gamma_{\mathrm{L}}}\right)\right], \tag{22}$$

where

$$\Omega = \sqrt{\omega_{\mathrm{r}}^2 - \gamma_{\mathrm{L}}^2}. \tag{23}$$

In this implementation, the model has five adjustable parameters: $\omega_{\mathrm{pl,D}}$, $\omega_{\mathrm{pl,L}}$, $\gamma_{\mathrm{D}}$, $\gamma_{\mathrm{L}}$ and $\omega_{\mathrm{r}}$. The susceptibilities of unperturbed samples, $\chi_\omega^{(0)}$, were determined from separate field-resolved measurements, where the pump pulse was blocked and test waveforms were recorded with and without the sample. We then used equation (11) to evaluate $\chi_\omega^{(0)}$ and the sample thickness.

The reconstruction results that we present in the main text were obtained by numerically optimizing the fit parameters, for which we minimized the difference between the measured and reconstructed waveform distortions. The reconstructed waveforms were evaluated with the aid of equations (10), (12), (13), (17) and (20)–(23). Extended Data Table 1 lists the obtained values of the fit parameters.

### Nonlinear work
In Fig. 1b, we show the time-dependent nonlinear work (green curve) as a means for illustrating the dynamics of photoinjection. We obtained these data using the SALMON code[28], which implements the real-space real-time time-dependent density functional theory. Using the Tran–Blaha meta-generalized-gradient-approximation exchange potential with Perdew–Wang correlation[41], we calculated the macroscopic electric current density, $J(t)$, induced by a realistic laser pulse shown in Extended Data Fig. 1. The peak electric field of the pulse inside the silicon crystal was set to 0.35 V Å$^{-1}$. The time-dependent work performed by the electric field $E(t)$ is given by $W(t) = \int_{-\infty}^{t} E(t')J(t')\mathrm{d}t'$. To get the nonlinear work, we ran another simulation with a much

weaker pulse (0.001 V Å$^{-1}$), which gave us a good estimation of the linear work. In the absence of single-photon transitions, the linear work is responsible for the linear polarization that the pulse transiently induces in the medium. The linear work scales as the square of the peak electric field, which allows one to calculate the linear component of the work performed by an intense pulse. By subtracting the linear work from the total work, we obtained the nonlinear component of the total work, which is due to nonlinear interband transitions and, to a lesser extent, the motion of photoinjected charge carriers.

### Optical system
A Ti:sapphire oscillator (Rainbow 2, Spectra Physics) was used to provide an octave-spanning bandwidth, centred at 750 nm (Extended Data Fig. 2). The carrier-envelope-phase (CEP) from the oscillator was stabilized using a feed-forward scheme. The pulses were further amplified within a nine-pass cryo-cooled Ti:sapphire chirped-pulse amplifier at a repetition rate of 3 kHz and temporally compressed using a transmission grating-based compressor, yielding 21 fs pulses with 2.5 W output power. The amplified pulses were spectrally broadened in a hollow-core fibre filled with neon gas (1.8 bar pressure), resulting in a spectral broadening that covered the wavelength range from 400 nm to 1,100 nm. Three pairs of chirped mirrors in combination with 6 mm of fused silica (and 50 cm of air) were used to compress the pulses to about 3 fs (full-width half-maximum of the intensity envelope, evaluated from nonlinear photoconductive sampling[3] measurements). The compressed pulses were then guided to the experimental setup for further experiments. An extra CEP stabilization loop, for long-term drifts, was implemented using the fibre output and the $f$-to-$2f$ technique.

### Data acquisition
The electric-field waveforms were recorded with a recently demonstrated GHOST[29] with a heterodyne signal produced in a $z$-cut $\alpha$-quartz crystal of about 12.3 µm thickness. The optical signal for GHOST detection was measured with a fast photodiode (Roither Lasertechnik). A transimpedance amplifier (DLPCA-200, FEMTO Messtechnik) was used to provide an amplified voltage signal to a gated integrator (SR250 Stanford Research Systems) that was triggered by an electrical signal synchronized to the repetition rate of the laser. The integrator produced a d.c. voltage that was proportional to the integrated photodiode signal for all combinations of optical pulses (see the next section). The signal from the integrator was recorded using an oscilloscope (Tektronix) for analogue-to-digital conversion and recorded using a computer interface to the General Purpose Interface Bus (National Instruments).

### Multiplexed detection of perturbed and unperturbed waveforms
Fluctuations and drifts in the laser system modify the test pulse during a delay scan, which presents a major obstacle for measuring small changes in electric-field waveforms. To mitigate this problem, we measured the modified and reference waveforms almost simultaneously. For each arrival time of the sampling pulse, we measured the following three signals: (1) a signal generated by the sampling pulse interacting with the modified waveform; (2) a signal for which the sampling pulse interacted with the reference waveform and (3) a signal for which only the sampling pulse reached the photodetector. This form of detection was implemented using optical chopper wheels to transmit every third pump and block every third test pulse, generating the following sequence of pulses:

| | | | |
|---|---|---|---|
| Pump pulse: | 1 | 0 | 0 |
| Test pulse: | 1 | 1 | 0 |
| Sampling pulse: | 1 | 1 | 1 |

Here '1' indicates that the pulse is present, whereas '0' indicates that the pulse is blocked by the chopper wheel.

This sequence of pulses is delivered to the photodiode. The current signal from the photodiode is then further amplified and converted to a voltage by means of a transimpedance amplifier (DLPCA-200, FEMTO Messtechnik). The output from a transimpedance amplifier provides a time-domain train of voltage signals (Extended Data Fig. 3a). A boxcar integrator (SR200, Stanford Research Systems) provides a d.c. output for each of these voltage pulses (Extended Data Fig. 3b): $d.c._\alpha$, $d.c._\beta$ and $d.c._\gamma$ correspond to the three columns in the above table, respectively. The output from the boxcar integrator is digitized and stored. The pump pulse is blocked after the sample and does not reach the detector.

If the pump pulse does not change any properties of the sample medium, the $d.c._\alpha$ and $d.c._\beta$ outputs are identical. If, however, the pump pulse changes the sample medium, then the difference between $d.c._\alpha$ and $d.c._\beta$ outputs corresponds to a difference in the test waveform due to the change of the sample properties. The signal related to the modified waveform is $d.c._\alpha - d.c._\gamma$, whereas the signal related to the reference waveform is $d.c._\beta - d.c._\gamma$.

## The experimental setup

The optical setup is shown in Extended Data Fig. 4. The pump pulse, obtained from the reflection off the front surface of the first wedge in the wedge pair (WP 1), was delayed using a closed-loop piezo stage (PX 200, Piezosystems Jena). The sampling pulse is obtained by taking the reflection from a 200-mm-thick ultraviolet-grade fused silica window (FS 2). The reflected light was delayed using another closed-loop piezo stage (PX 200, Piezosystems Jena). Wire-grid polarizers (WG 1, WG 2, WG 3 and WG 5) were used to control the energies of the test, pump and sampling pulses, whereas fused-silica wedge pairs (WP 1, WP 2, WP 3) were used for fine dispersion compensation and CEP control. An off-axis protected silver parabolic mirror (OPM 2) was used to focus the test arm on a 0.8-mm-thick ($\theta = 10.3°$, where $\theta$ is the angle between the surface normal and the crystal axis) BiBo crystal (NL 1) for intra-pulse difference-frequency generation. A set of fused-silica and silicon plates (FS 1, Si) was used for the compression of the near-infrared pulses and for blocking the fundamental, respectively. The near-infrared light was collimated by a protected gold off-axis parabolic mirror (OPM 3).

The pump and test pulses were recombined using a wire-grid polarizer (WG 4), which transmitted the pump pulses and reflected the test pulses. The orthogonally polarized pulses were focused using a protected silver off-axis parabolic mirror (OPM 1) on the sample and re-collimated using another protected silver off-axis parabolic mirror (OPM 4). The pump pulses were transmitted through a hole in the re-collimation mirror (OPM 4). The collimated beam was recombined with the sampling pulses using a wire-grid polarizer (WG 6). A 12.34-mm-thick $z$-cut quartz crystal (NL 2) upconverted the sampling pulse. After spectral filtering by a bandpass filter (BP 1) and polarization control by a wire-grid polarizer (WG 7), this high-frequency light served as the local oscillator for generalized heterodyne optical sampling. At the same time, NL 2 enabled nonlinear wave mixing between the sampling and test pulses. The interference of this signal with the local oscillator was detected with a SiC-based photodiode (PD 1). Modulation of the test and pump pulses was accomplished using chopper wheels (CW 1 and CW 2), generating the sequence of pulses for the almost simultaneous waveform detection.

## Timing of the injection event

The following experimental procedure was used to determine the moment of photoinjection (delay zero). The experimental setup has two focal planes: the sampling and detection ones. The pump and test pulses interact with a thin sample in the sample plane. Before the transmitted test pulse reaches the detection plane, the transmitted pump pulse is removed and a sampling pulse is added to the beam. GHOST detection then takes place in the detection plane, where the two pulses interact with the nonlinear crystal that is labelled as NL 2 in Extended Data Fig. 4. The wedge pair WP 2 was used to set the CEP of the pump pulse in the sample plane to 0 (cosine pulse). The wedge pair WP 3 was used to set the CEP of the sampling pulse in the detection plane to 0. The pump pulse in the sample plane is a replica of the sampling pulse in the detection plane because both pulses are derived from the same incoming pulse, they have the same CEP and they propagate through the wedges made of the same medium under conditions adjusted for maximum compression.

In the detection plane, the strong sampling pulse produces a nonlinear gate confined to the vicinity of the strongest peak of the sampling pulse. The gate was scanned through the test pulse by a retroreflector consisting of mirrors M6 and M7 placed on piezo translation stage TS 2. As the nonlinear gate is confined to the vicinity of the strongest peak of the sampling pulse, the signal produced at each position of the TS 2 corresponds to the temporal overlap between the strongest peak of the sampling pulse and the part of the test pulse within the window of the duration of the gate (less than 1 fs). Hence, by sampling the test-pulse waveform, one can the map positions of translation stage TS 2 to temporal overlaps of the peak of the sampling pulse with various parts of the test waveform.

To determine the moment of photoinjection in the sample plane (delay zero with respect to the test pulse), the sample was removed and a nonlinear crystal NL 2 was installed in the sample plane. Then the test pulse was measured in the sample plane using the same GHOST scheme as in the detection plane but with the pump pulse playing the role of the sampling pulse. This not only allows one to characterize the test pulse in the sample plane but also to map the positions of the translation stage TS 1 to temporal overlaps of the peak of the pump pulse with various parts of the test pulse. Extended Data Fig. 5 shows an example of such a pair of measurements.

After measuring the test-pulse waveform in both sampling and detection planes, the relation between the positions of translation stages TS 1 and TS 2 can be determined. Hence, various moments of photoinjection in the sample plane can be projected onto the test-pulse waveform measured in the detection plane. When the injection translation stage is moved to position $x$, this corresponds to the excitation of the sample at the location on the test-pulse waveform that corresponds to the position of the translation stage in the detection plane being $y = x + d$.

The constant shift $d$ was measured before each experimental campaign by sampling the test pulse in the sample and detection planes with the same optics and electronics. Once $d$ is determined, the moment of the photoinjection can be mapped directly on the measured test-pulse waveform in the detection plane.

## Data availability

The measured and simulated data for all the figures in this paper are available from the Edmond Data Repository of the Max Planck Society[42]. Source data are provided with this paper.

## Code availability

The codes used to reconstruct the measured waveform distortions and prepare the plots are available from the Edmond Data Repository of the Max Planck Society[42].

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

**Acknowledgements** We thank S. Sederberg and M. Uemoto for discussions. This research is based upon work supported by the Munich Centre for Advanced Photonics. N.K. was partially supported by the FISR-CNR project 'TECNOMED—Tecnopolo di nanotecnologia e fotonica per

la medicina di precision'. M.Q. and D.A.Z. acknowledge support from the Max Planck Society via the International Max Planck Research School for Advanced Photon Science (IMPRS-APS).

**Author contributions** F.K. and N.K. conceived the project. D.A.Z. performed the experiments. D.A.Z., M.Q. and V.S.Y. analysed the experimental data and performed the theoretical modelling and calculations. All authors discussed the results and contributed to writing the manuscript.

**Funding** Open access funding provided by the Max Planck Society.

**Competing interests** The athors declare no competing interests.

**Additional information**
**Correspondence and requests for materials** should be addressed to Nicholas Karpowicz or Vladislav S. Yakovlev.

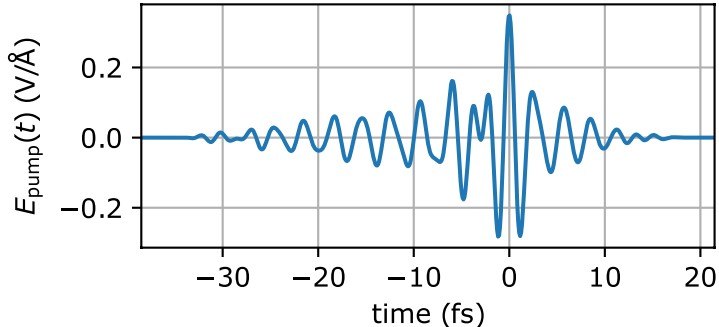

**Extended Data Fig. 1 | The electric field of the pump pulse that was used in the TDDFT simulations.** Figure 1b in the main text shows the central part of this waveform, which was obtained by first measuring the pump pulse with the aid of nonlinear photoconductive sampling and then adapting these measurement results for numerical simulations: suppressing the wings and removing the noise by spectral filtering.

Ti:Sa oscillator
70 MHz
~ 1.2 nJ
~ 6 fs

Stage I
CEP stabilization

Ti:Sa CPA
3KHz
~ 950 uJ
~ 21 fs

Chirped mirror
compressor
~ 400 uJ
~ 2.7 fs

Hollow core fiber
Ne @ 1.8 bar

Pointing
stabilization

To experiment

Stage II
CEP stabilization

**Extended Data Fig. 2 | Diagram of the laser source layout.** The output of the Ti:Sa oscillator is guided through the CEP4 module (Femtolasers) for CEP stabilization (Stage I carrier-envelope phase stabilization). The transmitted pulse is amplified in a chirped-pulse amplifier (Ti:CPA). The amplified pulse is guided into the hollow core fiber filled with neon gas. The beam is coupled into the fiber following active stabilization. After the hollow core fiber, the beam passes through three pairs of chirped mirrors (Ultrafast Innovations) which compensate for the spectral phase of the incident pulse as well as the additional dispersion of the experimental setup. In front of the compressor, an outcoupling mirror reflects a small fraction of the beam into the second stage of the carrier-envelope phase stabilization, which is based on the f-2f scheme.

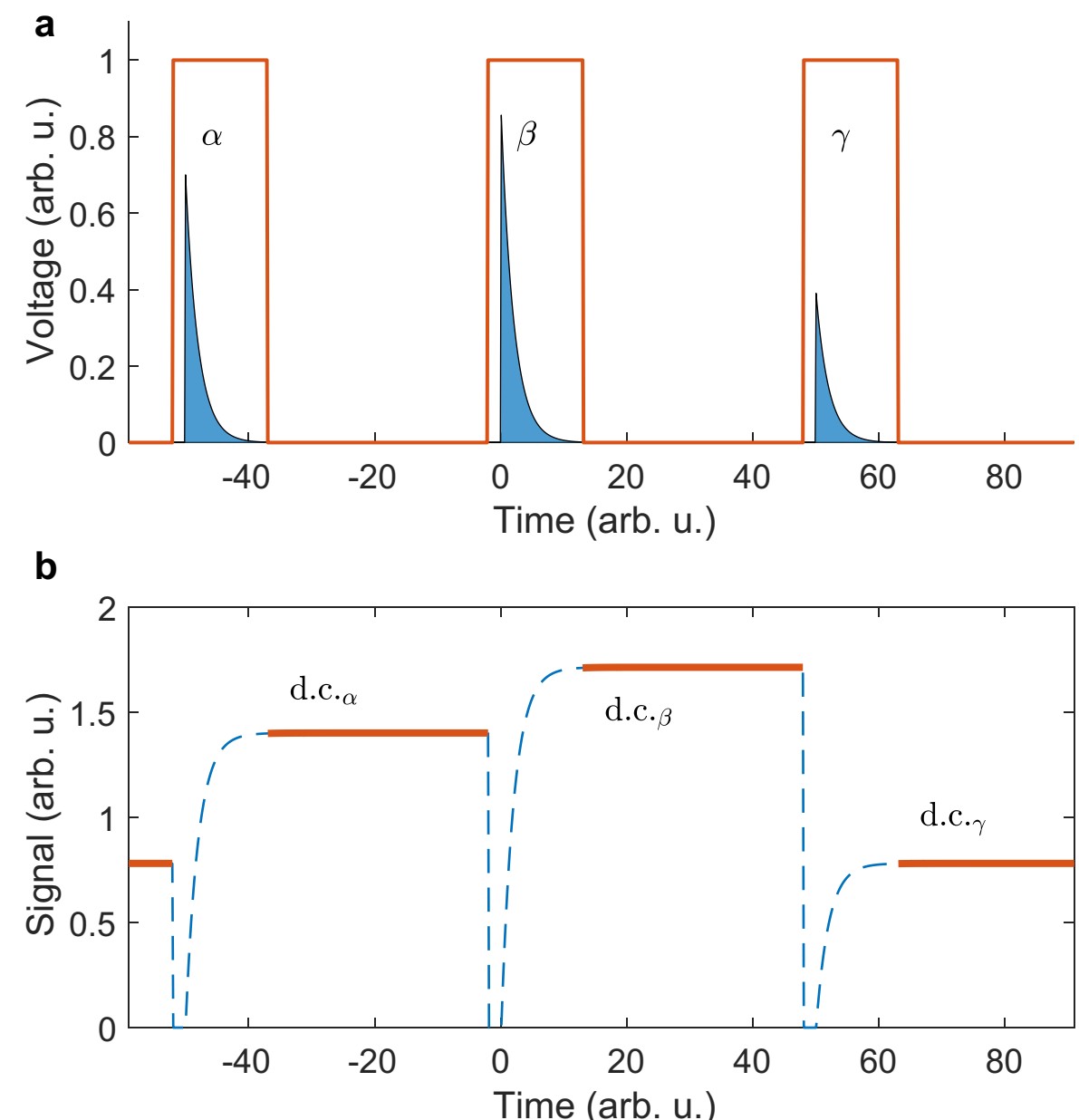

**a**

**b**

**Extended Data Fig. 3 | Schematic representation of the multiplexing scheme. a**, A photodiode registers three consecutive optical pulses labeled as α, β, and γ. Each pulse produces an electrical signal with the shape of a decaying exponential function (blue). The integral of this signal is proportional to the energy carried by the respective optical pulse. The orange outlines represent the windows of the integration time. **b**, The boxcar integrator generates an output d.c. signal after each integration window. This d.c. value can be further read out with an oscilloscope or any analog-to-digital converter device.

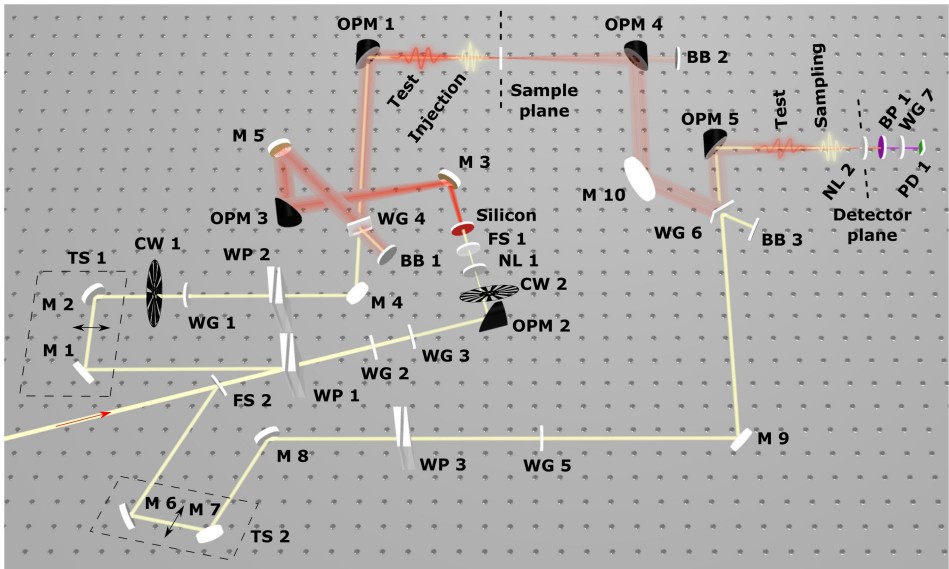

**Extended Data Fig. 4 | Schematic of the experimental setup.** An optical pulse from the laser source is split into three optical arms: test, injection and sampling. The pulse in the test arm is down-converted to a pulse with a central wavelength of 2.1 μm. The test and injection pulses are combined with a controlled delay between them and transmitted through a sample. The injection pulse is then filtered out, while the transmitted test pulse is combined with the sampling pulse, which nonlinearly interacts with the test pulse in the detection plane, sampling its electric field. This figure was created with the online 3Doptix platform (https://3doptix.com/).

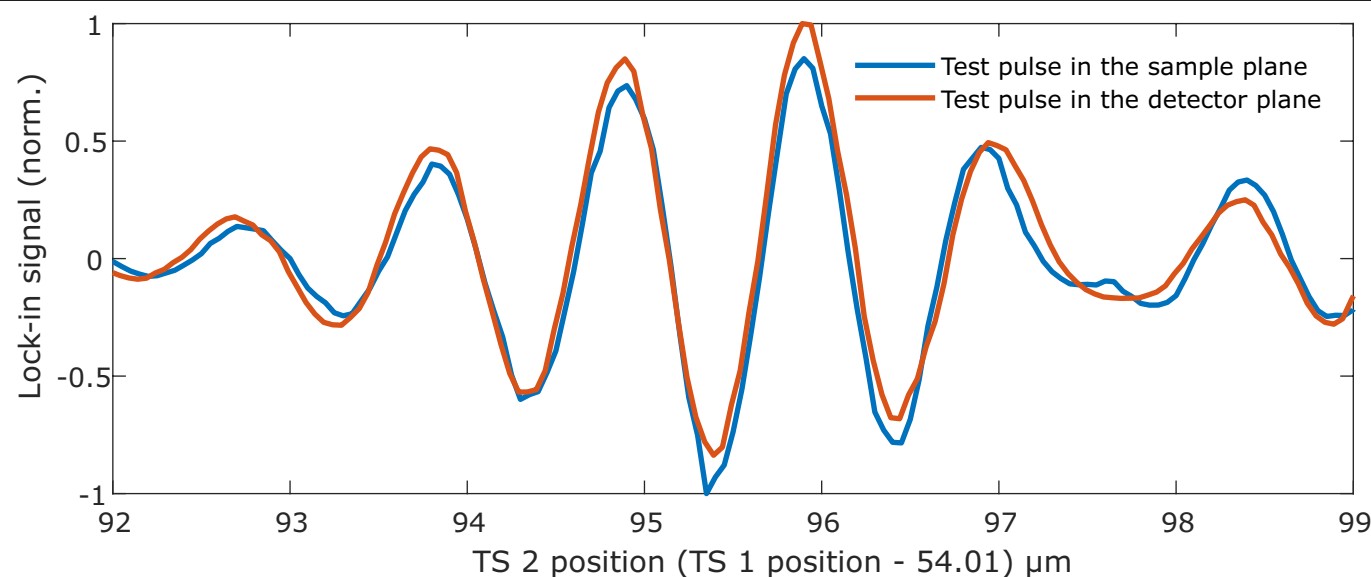

**Extended Data Fig. 5 | Timing (translation stage positions) synchronization between two focal planes.** Since photoinjection takes place in the sample plane, while the test pulse is sampled in the detection plane, it is necessary to calibrate the delay between the pump and test pulses. This is done by determining the mapping between the positions of piezo stages that control the pump-test and sampling-test delays, which is accomplished by comparing test waveforms measured in the sample and detection planes.

**Extended Data Table 1 | The retrieved values of the fit parameters**

| Parameter | Silicon | Fused silica |
|---|---|---|
| $\omega_{\mathrm{pl,D}}$ (fs$^{-1}$) | 2.30 | 0.0566 |
| $\gamma_{\mathrm{D}}$ (fs$^{-1}$) | 0.494 | 0.945 |
| $\omega_{\mathrm{pl,L}}$ (fs$^{-1}$) | 0.629 | 0.0282 |
| $\gamma_{\mathrm{L}}$ (fs$^{-1}$) | 0.0155 | 0.0592 |
| $\omega_{\mathrm{r}}$ (fs$^{-1}$) | 0.716 | 0.344 |

The values of the parameters of the Drude–Lorentz model (see Methods) that minimize the discrepancies between the measured and simulated waveform distortions.