## [Peer Review File · Nature]

Manuscript Title: Dynamic optical response of solids following 1-fs-scale photoinjection

Reviewer Comments & Author Rebuttals

Reviewer Reports on the Initial Version:

Referees' comments:

Referee #1 (Remarks to the Author):

Manuscript ID: 2021-12-20463

Title: "1-femtosecond-scale Drude-Lorentz response in photoexcited solids"

D. A. Zimin and coworkers present an interesting experimental work which deals with a timing and interesting topic: ultrafast motion/creation of charges under strong non-equilibrium conditions in solids. Despite the huge amount of recent experimental and theoretical work on this matter, the physical phenomena underlying light-matter interaction in these regimes are far from being fully understood. The approach used by the authors, namely to observe the transient changes induced by ultrafast carrier injection over a reference waveform, represents a nice alternative to pump-probe experiment based on attosecond pulses (which I found poorly referenced through the work). By comparing the experimental results with theoretical calculations based on the Drude-Lorentz model, instantaneous electron excitation and linear polarization, the authors find subtle differences which become important around the first few femtoseconds during the interaction between the sample and the pump pulse. At later times, the sample response seems to follow the model. Based on this, the authors conclude that the Drude-Lorentz response forms within few femtoseconds despite the plasma frequency would predict a slower behavior (at least in silica).

The work is well written and presented. The treated topic is relevant. Nevertheless, I have serious doubts on how the physical mechanisms happening during light-matter-interaction have been modeled and the connection of the present results with the existing literature (see my detailed points below). As the silicon results are compatible with the timing dictated by the plasma frequency, the main finding of the work stems from the silica data. In silica, though, the results show that the system non-linear response is confined only in the time frame of the light-matter interaction. To my understanding this has already been proven from the same group in 2013 (M. Schultze et al., *Nature* **493**, 75-78 (2013)). Therefore, I cannot recommend publication on *Nature*, or any other journal, before the authors provide a convincing and solid answer.

Major comments:

Both in Si and SiO₂ the authors assume carrier injection to happen by multi-photon absorption. Under this hypothesis, the carrier population follows in time the n-th power of the field intensity, where n is the number of photons simultaneously absorbed. The higher the number of photons required to reach the conduction band, the shorter the injection mechanism in time, which justifies the assumption of sudden charge creation upon which the theoretical interpretation of the work is based. Nevertheless, I doubt that multiphoton-absorption and linear polarization can

describe light-matter interaction in both cases, thus compromising the key elements upon which the description and interpretation of the reported results are based. I here explain my reasons in the two cases.

Silicon.

According to the authors Si carriers are supposed to be injected into the conduction band by a two-photon absorption process, following the square of the pump field intensity as reported in Fig. 1b. I have serious doubts that two-photon absorption is the right mechanism to describe carrier injection. In a seminal and closely-related paper not cited by the authors (M. Schultze et al., *Science* **346**, 1348-1352 (2014)), Schultze and coworkers have shown that carrier injection in Si deviates from two-photon absorption already at field peak intensities inside the solid of about $5\text{-}6 \times 10^{11} \text{ W/cm}^2$ (see figure 3D in the work from Schultze). With the field strength given by the authors in the manuscript, $E = 0.8 \text{ V/\AA}$, and assuming a central wavelength of 750 nm as reported in the supplementary information (so that $n_{\text{Si}}(750\text{nm}) = 3.7$), we get a peak intensity of $I_{\text{peak}} = 2I_{\text{average}} = \epsilon_0 n_{\text{Si}}^2 c E^2 \cong 6 \times 10^{13} \text{ W/cm}^2$ which surely does not justify the assumption of two-photon absorption, but rather seems to suggest that the main injection mechanism is tunnelling. To remove any doubt over my calculation of the peak intensity, we can compare directly the field amplitude, $E = 0.8 \text{ V/\AA}$, with the one reported by Schultze in his work at figure 3B. While the amplitude in the work from Schultze never passes 0.6 V/\AA , they find that the main injection mechanism is tunnel ionization, which happens at each half-cycle during the interaction and does not follow the law presented in Fig. 1b. Even if the charge injection may be anyway confined around the field maximum, this proves that the system is not in the linear regime.

Silica.

It is undoubted that energy balance requires more photons to be absorbed in order to promote an electron from the valence band to the conduction band in this case. Nevertheless, as the authors write, due to the relatively the strong field used in the experiment, $E = 1.4 \text{ V/\AA}$, the Keldysh parameter is close to 1. Multiphoton absorption is, instead, expected to happen for a Keldysh parameter $\gg 1$ (H. Mashiko et al., *Nat. Commun* 1468 (2018)).

The field strength used by the authors is similar to what used by some of them in another work: A. Schiffrin et al., *Nature* **439**, 70-74 (2013), not cited here. Schiffrin showed that important strong-field mechanism, here neglected, influence light-matter interaction and carrier injection in SiO₂. These phenomena do not necessarily follow the n-th power of the field intensity as for the case of multiphoton absorption. The charge due to the field induced polarization (Fig. 2b) seems to scale exponentially with the applied field strength (in the same range as for the present work).

In another related work from the authors themselves: M. Schultze et al., *Nature* **493**, 75-78 (2013), not cited in the manuscript, the authors have already shown one of the main results of the current manuscript as they found that for silica: *“Both the nonlinear polarization and the conduction band population induced by the strong field return to near-zero immediately after the laser pulse for $F \leq 2.5 \text{ V/\AA}$. This is fully consistent with the abrupt decay of field-induced transient NIR reflectivity and XUV absorption bleaching: see Figs 3b and 2b, respectively. These results imply that the sample exposed to fields as high as $F \leq 2.5 \text{ V/\AA}$ resumes its original (field-free) state immediately after exposure.”*

Therefore, it is not too surprising that the Drude-Lorentz model can catch the system response in

SiO₂ soon after the interaction with the pulse.

Finally, in the work from Sommer et al. (Ref. 14 of the present work), the authors show that the nonlinear polarization in silica can have important effects, inducing an energy transfer between the field and the material which lasts for all the pulse duration and it is not confined in the main pulse cycle.

The theoretical model used to fit the data assumes linear polarization and sudden ionization. Based on what written above, I believe these to approximations to be unjustified. While one can state that the non-linear polarization and the finite ionization time window are responsible mainly for the deviation between the model and the data observed in Fig. 4 around 0 fs, I wonder if the agreement at big positive and negative time delays is robust. As the authors write in the supplementary information the model has 5 free parameters, which gives quite some room in terms of fitting capability. How robust is the fit? Or, in other words, can the fit reproduce any experimental result?

The authors write *“... we evaluate an upper limit for the formation time of the Drude-Lorentz response as 4 fs—there is no systematic deviation between the measurement and the Drude-Lorentz fit for $t > \tau + 4 \text{ fs}$ ”*. If we neglect the comments at point 1 and 2 above and try to evaluate the upper limit for the for the formation time of the Drude-Lorentz response, I believe that we should start counting from the time the system displays a non-zero response (interaction with the laser) and not from $t = \tau$ as figure 3f suggests. Here we see that the system response starts roughly 5-6 fs before the black dashed line. Therefore, the total time it takes to reach Drude-Lorentz would be about 10 fs at least.

The existence of a non-zero signal for some femtoseconds before the black-dashed line in Fig. 3f shows that the carrier injection is not instantaneous in SiO₂. Otherwise, how can the probe waveform be modified if the carriers are yet to be created?

Minor comments:

Since the model cannot reproduce what happens during light-matter interaction and since the carrier injection mechanism is not clear, I find the use of 1-fs-scale potentially misleading. I suggest to remove it.

It is not clear what the 2.8 fs refers to in Fig. 1b. Is it the supposed window in which the carrier injection takes place? How has this been defined?

While the signal in fig. 4 reduces by an order of magnitude when moving from Si to SiO₂, the error bars stay comparable. Is it due to better statistics for the SiO₂ case?

In figure 4 a legend for the open circles and squares is missing.

The reference style seems not to be uniform. Titles are missing.

The work from Sommer on the nonlinear polarization is referenced while talking about instantaneous ionization in the introduction. While I believe the work to be important and to be cited, I do not see the direct link.

There is a typo at line 499 when indicating the equation used to evaluate the reconstructed waveforms. Is it possible that “((21)-(23))” should instead be “(20)-(22)”?

Referee #2 (Remarks to the Author):

The paper reports an sub-cycle-resolved observation of how mid-infrared (MIR) optical field response to the photoexcited solids, silicon and fused silica, during the first few femtoseconds after the instantaneous photo-injection of carriers which is temporally confined to a single dominant half cycle of the electric field. Based on the observation of the modification of the MIR probe electric field from the reference one, the authors concludes that the Drude-Lorentz response forms within a few femtoseconds after the photo-injection, in particular, this time was much shorter than the inverse plasma frequency in the case of fused silica.

In my opinion, there are two important aspects in this manuscript. One is the authors' excellent problem setting to be solved in this work. The fundamental question how fast electron-hole plasma forms in photoexcited solid state materials has been a focus of attention in not only the field of attosecond physics but also solid state physics. This is because the time required electron-hole plasma formation after the initial photoexcitation is closely related to the quantum many-body dynamics in which charge screening play an essential role. Clarifying the dynamics and characteristic time scale of this plasma formation or charge screening is still challenge to ab initio approach. From not only theoretical but experimental point of view, this is not easy task. Generally, it is predicted to be comparable to the inverse plasma frequency, which corresponds to the time scale of hundreds of attoseconds in metals and tens of femtoseconds in semiconductors. To date, one famous experimental work, which is referred as Ref. 7 in this manuscript, was reported for investigating GaAs semiconductor over 20 years ago. Although the charge screening dynamics has been long-standing mystery, it is still undoubtedly one of the deepest and the most fundamental research targets as an unestablished many-body dynamics in the field solid state physics.

The other important aspect is the authors' advanced spectroscopic technique named as generalized heterodyne optical sampling technique (GHOST). This technique enables us to directly measure the electric field of light in time domain with very simple experimental setup compared to conventional optical field sampling technique such as attosecond streak method. By combining GHOST to pump-probe spectroscopy, the authors achieved the direct measurement of the probe electric field passed through the photo-injected materials. Electric field measurement provides us optical response of real and imaginary part, which correspond to refractive index and absorption coefficient of photo-injected solid state materials

Thus, experimental approach of the electron-hole plasma formation dynamics will be one of the exciting, broad-interest and important targets in the attosecond science. In addition, the carrier screening dynamics will interest many researchers in broad field of solid state physics. Therefore, the scientific importance and impact described in this manuscript is in the scope of Nature. However, I find that their main achievements in the current manuscript do not contain enough sufficient interpretation of experimental data for justifying a publication in Nature.

The main conclusion of the authors is derived from the experimental observation that the probe MIR electric field is modified with the photo-injection in Si and SiO₂ sample. My main concern is that it is unclear how the authors distinguish the electron-hole plasma state (collective response under the screened Coulomb collisions) from the free carrier state where unscreened bare Coulomb collisions is dominant in their experimental data. In the following I describe my main opinions and concerns in detail.

(1) In my knowledge, the first experimental achievement which clearly showed electron-hole plasma

formation in time domain was Ref. 7 (Huber et al., Nature 414, 286 (2001).) They used THz single-cycle electric field as a probe and 100-fs near infrared pulse as a photo-injection pulse, and then investigated electron-hole plasma formation in photoexcited GaAs semiconductor. The experimental concept is very similar with the experimental setup in the current manuscript. In Ref. 7, they measured the dielectric function of the photoexcited GaAs by the THz probe electric field, and concluded the electron-hole plasma formation by confirming the appearance of additional oscillatory signal which originates from the electron-plasma formation. This was supported by the spectral peak of the imaginary part of the inverse dielectric function obtained by the signal matches the estimated plasma frequency. They also confirmed the real part of the inverse dielectric function well described a dressed Coulomb interaction, which is characteristic of electron-hole plasma formation (Fig. 3 (a) & (b) in Ref. 7).

In contrast, the authors did not analyze the modified probe electric field of MIR for obtaining the dielectric function of the photo-injected sample. In analogy to the analysis described in Ref. 7, the modified probe electric field (MIR waveform distortion) must have information of the plasma formation, such as plasma frequency. In Si, the authors describes the plasma period to be 2 – 3 fs, which corresponds to roughly 400 THz. It is unclear why the data shown in the current manuscript does not show any sign for the plasma formation like the data in Fig. 3 in Ref. 7. The differences between the current manuscript and Ref. 7 are (i) the probe wavelength (2.2 micron in this study and THz in Ref. 7), (ii) the sample (silicon and silica in this study and GaAs in Ref. 7), (iii) the intensity of the photo-injection pump pulse (two or five photon excitation in this study and resonance excitation in Ref. 7), and (iv) time scale for measurement (a few femtosecond or less in this study and a few tens of femtosecond to 200 fs in Ref. 7). If you can not distinguish the plasma state from the quasi-free state, it may be difficult to discuss the dynamics of electron-plasma formation and its dynamics. The authors should comment on this point or add a discussion in the revised manuscript.

(2) It seems that the description about the plasma frequency or plasma density is very few, which will support the authors' data interpretation. According to the "Methods" section, the plasma frequency in the Drude and Lorentz model is a fitting parameter. It is likely to be better for readers to get the information of plasma frequency or plasma density which is directly linked to its plasma frequency. In the case of Si, is the plasma frequency on the order of 10^{14} Hz? It is generally much difficult to estimate nonlinear excitation process compared to one-photon resonance excitation of electrons in the conduction band. Estimated plasma density, which can be derived from the plasma frequency, should be comparable to the estimated value from the photoexcited carrier density via nonlinear excitation. The authors should offer the information of plasma frequency to the readers, and discuss about it.

(3) When you compare the amplitude decrease in Fig. 4 (a) with that in Fig. 4 (b), the scale of the $\text{Im}[n\omega(t)]$ changes from about 0.8 to 1×10^{-3} . This difference must be due to the sample difference. Since $\text{Im}[n\omega(t)]$ is related to the photoexcited carrier density, it is nearly 3-order different between Si and silica. However, the rise time of Si (Fig. 4 (a)) is almost same as that of silica (Fig. 4 (b)), which may be 3 fs. Could you comment on the reason? If highly nonlinear photoexcitation occurs, the rise time of the photoexcited carrier density must be faster. Is the probe temporal resolution not enough for measuring such rise time?

(4) Fig. 4 (a) shows the onset of the change of the data shown in the upper and lower graphs

significantly different. The onset of the amplitude decrease represented by red circle starts about 4 fs earlier than that of the zero-crossing shift represented by blue circle. The slope of the zero-crossing shift is much steeper than that of the amplitude decrease. This tendency is similar in Fig. 4 (b). If the carrier excitation and the phase velocity change are induced by two-photon absorption and optical Kerr effect respectively, the onset and the slope of both curves should be almost identical. The authors can add the discussion about this discrepancy for readers.

(5) You can recognize a large deviation from the theoretical curve in the delay time from -7 to 0 fs in the lower graph of Fig. 4 (b). It looks very large deviation compared to other differences between experiments and theory shown in Fig. 4. The authors should comment on this point.

In summary, my major concern is that I don't see the clear experimental evidence which you can judge the formation of electron-hole plasma, compared to the previous experiment in Ref. 7. Since the novelty and impact of the manuscript depends on the question how quickly the electron-hole plasma formation when the photo-injection confined to sub-cycle time scale, the identifying the plasma formation experimentally is an inevitable problem to be solved for the measurement and the data analysis for satisfying the criteria of Nature publication in my opinion.

Referee #3 (Remarks to the Author):

In their work, Zimin and coworkers are investigating the response of bulk materials to ultra-short, femtosecond-scale excitations. They report experimental measurement of the change of the optical properties of silicon and fused silica following this excitation. These changes are analyzed in terms of a Drude-Lorentz response, and it is shown that the this response forms much faster than the inverse plasma frequency.

This work is clearly interesting and timely, and also well written. In particular, it allows for distinguishing the difference between the Drude-Lorentz response and the remaining part of the changes, the latter been qualitatively different for silicon and fused silica.

However, while the reported results are quite interesting, the referee finds that the paper is sometimes missing some deeper understanding and analysis of the reported results.

For instance, in Fig. 3, silicon and fused silica are behaving qualitatively differently for the non-Drude-Lorentz part of the waveform distortion. It would be good is the authors could give a tentative explanation why this is the case.

At the moment, it is stated that the observed changes in the central region Fig. 3F is related to the plasma period in fused silica. The reader is therefore left with questions like: Why isn't it also the case for silicon where the plasma period is between 2-3fs according to the manuscript, and should therefore be detectable? As the samples been of different thicknesses, could this originate from propagation effects?

Similar lake of understanding is for the Fig. 4 where the authors also note that "We cannot fully explain the magnitude of this shift" but associate this with possible propagation effects. Here, the paper would have really benefited by having measurement performed for different thicknesses, to be able to bring deeper understand of the reported effects.

The referee thinks that the results of the Drude-Lorentz should be better analysed and presented. Indeed, it would be interesting to get what are the obtained values after the fit, in particular for the plasma frequencies, and the resulting number of carriers N_D and N_L , to see if the extracted values are meaningful or not. The same is true for the lifetimes γ_L and γ_D . Moreover, why these plasma frequencies are different? Indeed, if one only consider one type of Lorentz oscillators, N_L should correspond to N_D . This is somehow directly implied by the sentence (lines 468-470) "once an electron makes a transition from a valence state to a state in a conduction band, the electron can then be further excited into a higher conduction band by absorbing a photon, while the vacancy left in the valence band can be filled by photoexciting an electron from a deeper valence band".

As a summary, while the presented results are certainly of interest, the referee finds that the present manuscript does not provide a deep understanding of the reported data, and thus seems to be suited for a publication in Nature.

Author Rebuttals to Initial Comments:

Response to reviewer's comments

Referee #1

D. A. Zimin and coworkers present an interesting experimental work which deals with a timing and interesting topic: ultrafast motion/creation of charges under strong non-equilibrium conditions in solids. Despite the huge amount of recent experimental and theoretical work on this matter, the physical phenomena underlying light-matter interaction in these regimes are far from being fully understood. The approach used by the authors, namely to observe the transient changes induced by ultrafast carrier injection over a reference waveform, represents a nice alternative to pump-probe experiment based on attosecond pulses (which I found poorly referenced through the work).

We thank the referee for the positive assessment of our work. Indeed, our approach is complementary to that in experiments that utilize attosecond pulses of extreme ultraviolet (XUV) light. We would like to stress the word “complementary” because these two kinds of measurements address different physical processes. An XUV pulse predominantly excites electrons from inner atomic orbitals, while optical pulses are ideally suited for studying processes that involve transitions between and motion within valence and conduction bands. When an XUV pulse is used to trigger electron dynamics (rather than probe them), it is usually too weak to drive nonlinear processes, while our work is based on the nonlinear injection of charge carriers. When an XUV pulse is used as a probe, it provides information different from that of optical probing because there is no direct relationship between a medium's permittivity in the extreme-ultraviolet part of the spectrum and that in the visible domain. For these reasons, we cite a relatively small number of papers related to measurements that employed attosecond pulses. In the revised manuscript, we cite “Controlling dielectrics with the electric field of light” and “Attosecond band-gap dynamics in silicon” by Schultze *et al.*

By comparing the experimental results with theoretical calculations based on the Drude-Lorentz model, instantaneous electron excitation and linear polarization, the authors find subtle differences which become important around the first few femtoseconds during the interaction between the sample and the pump pulse.

Just to avoid potential misunderstandings, let us stress here that the focus of our manuscript is not on what happens *during* photoinjection but on what happens *afterward*. The Drude-Lorentz model is not supposed to describe the nonlinear wave mixing that occurs during the time when both the pump and test pulses interact with a solid.

At later times, the sample response seems to follow the model. Based on this, the authors conclude that the Drude-Lorentz response forms within few femtoseconds despite the plasma frequency would predict a slower behavior (at least in silica).

The work is well written and presented. The treated topic is relevant. Nevertheless, I have serious doubts on how the physical mechanisms happening during light-matter interaction have been modeled and the connection of the present results with the existing literature (see my detailed points below). As the silicon results are compatible with the timing dictated by the plasma frequency, the main finding of the work stems from the silica data. In silica, though, the results show that the system non-linear response is confined only in the time frame of the light-matter interaction. To my understanding this has already been proven from the same group in 2013 (M. Schultze

et al., Nature 493, 75-78 (2013)). Therefore, I cannot recommend publication on Nature, or any other journal, before the authors provide a convincing and solid answer.

We do not see any overlap between the central claims of our present manuscript and those of the paper that some of us published together with Martin Schultze in 2013. In that paper, we saw that a strong infrared pulse could have a significant effect on the absorption of an extreme-ultraviolet (XUV) pulse as long as the two pulses overlapped, while the charge carriers injected by the infrared pulse did not cause any significant effect on the XUV absorption for non-overlapping pulses. That experiment provided no information about how a weak optical pulse would accelerate charge carriers photoinjected by the intense pump pulse, and it did not provide any information about how quickly the optical response of silica to the test pulse would begin to match that of the Drude-Lorentz model.

Major comments:

Both in Si and SiO₂ the authors assume carrier injection to happen by multi-photon absorption. Under this hypothesis, the carrier population follows in time the n-th power of the field intensity, where n is the number of photons simultaneously absorbed. The higher the number of photons required to reach the conduction band, the shorter the injection mechanism in time, which justifies the assumption of sudden charge creation upon which the theoretical interpretation of the work is based.

Just to avoid any misunderstanding, let us point out here that the terms “multiphoton absorption” and “multiphoton ionization” have always been ambiguous: sometimes, they are used to say that an energy gap does not allow for single-photon absorption; sometimes, they are used to say that the nonlinear ionization cannot be viewed as a tunneling process. What matters is that our analysis of the experimental data is not based on any assumption regarding the mechanism of carrier injection except the assumption that most charge carriers emerge within a time interval that is shorter than half a cycle of the test pulse. This assumption follows from the fact that the pump pulse is absorbed nonlinearly.

Nevertheless, I doubt that multiphoton-absorption and linear polarization can describe light-matter interaction in both cases, thus compromising the key elements upon which the description and interpretation of the reported results are based. I here explain my reasons in the two cases.

Silicon.

According to the authors Si carriers are supposed to be injected into the conduction band by a two-photon absorption process, following the square of the pump field intensity as reported in Fig. 1b. I have serious doubts that two-photon absorption is the right mechanism to describe carrier injection. In a seminal and closely-related paper not cited by the authors (M. Schultze et al., Science 346, 1348-1352 (2014)), Schultze and coworkers have shown that carrier injection in Si deviates from two-photon absorption already at field peak intensities inside the solid of about $5\text{-}6 \times 10^{11} \text{ W/cm}^2$ (see figure 3D in the work from Schultze). With the field strength given by the authors in the manuscript, $E = 0.8 \text{ V/\AA}$, and assuming a central wavelength of 750 nm as reported in the supplementary information (so that $n_{\text{Si}}(750\text{nm}) = 3.7$), we get a peak intensity of $I_{\text{peak}} = 2I_{\text{average}} = \epsilon_0 n_{\text{Si}} c E^2 \cong 6 \times 10^{13} \text{ W/cm}^2$ which surely does not justify the assumption of two-photon absorption, but rather seems to suggest that the main injection mechanism is tunnelling. To remove any doubt over my calculation of the peak intensity, we can compare directly the field amplitude, $E = 0.8 \text{ V/\AA}$, with the one reported by Schultze in his work at figure 3B. While the amplitude in the work from Schultze never passes 0.6 V/\AA , they find that the main injection mechanism is tunnel

ionization, which happens at each half-cycle during the interaction and does not follow the law presented in Fig. 1b.

The referee is right that, for such a strong field, the nonlinear absorption of the pump pulse in silicon is not as simple as two-photon absorption. However, the goal of Fig. 1b is to give a reasonable estimation for photoinjection confinement in the simplest possible way. Prompted by the referee, we simulated the interaction of 3.5-fs 750-nm pulses with crystalline silicon using SALMON (<https://salmon-tddft.jp>), which is a code that implements the real-time real-space time-dependent density functional theory (TDDFT). The same code was used by Schultze *et al.* in *Science* **346**, 1348-1352 (2014). We estimate that the peak electric field of the pump pulse in the medium was 0.35 V/\AA (see our response to Referee #2). We found that the results change significantly depending on the exchange-correlation potential that we use. Schultze *et al.* used the local-density approximation (LDA), which underestimated the band gap. The fundamental band gap, in this case, is 0.56 eV, while the experimental value is 1.12 eV. For more realistic results, we also employed the Tran-Blaha meta-GGA exchange with Perdew-Wang correlation (TBmBJ), which gave us a fundamental band gap of 1.21 eV. The following plot shows how the concentration of photoinjected electrons depends on the peak electric field of the pump pulse:

As the referee pointed out, there are significant deviations from the two-photon scaling (dashed curves) at 0.35 V/\AA . However, in the context of our work (and Fig. 1b in particular), the essential question is the duration of the time interval during which most photoinjection occurs. If our simple estimation provides a reasonably accurate answer, there is no need to change Fig. 1b, and it would be better to keep the simple illustration of the concept. To address this question, let us consider the nonlinear work performed by the pump pulse on the medium. (The current version of SALMON does not allow one to evaluate time-dependent electron concentrations, and the very concept of instantaneous ionization probability in the presence of a strong electric field is disputed.) In the figure below, we show the nonlinear work for $E_0 = 0.35 \text{ V/\AA}$, highlighting the time interval between the moment when the nonlinear work reaches 10% of its final value for the first time and the moment when it reaches 90% of its final value for the last time. The duration of this time interval is shown in the plot legends.

For both exchange-correlation functionals, the TDDFT calculations give the photoinjection confinement that is tighter than the one predicted by the simplistic two-photon rate, which is 2.8 fs. Therefore, we would prefer to keep Fig. 1b as it is.

Even if the charge injection may be anyway confined around the field maximum, this proves that the system is not in the linear regime.

This comment signals that the referee may misunderstand why the polarization response to a sufficiently weak test pulse is always *linear*, no matter how nonlinear the interaction with the pump pulse is. Linearity has a precise mathematical definition. When the polarization response, $P(t)$, to an external electric field, $E(t)$, is linear, then there is a linear operator that relates these two functions. That is, the polarization response induced by $E(t) = cE_1(t) + E_2(t)$ is $P(t) = cP_1(t) + P_2(t)$ for any constant c and any choice of the two fields, where $E_1(t)$ alone would induce $P_1(t)$, while $E_2(t)$ alone would induce $P_2(t)$. Note that this is a very general definition that allows the properties of the medium to change with time. Even if the interaction of the medium with a pump pulse is extremely nonlinear and the pulse overlaps with a test pulse, the part of the polarization response that emerges from the interaction with an infinitesimally weak test pulse will satisfy the above requirements. This is obvious in the conventional nonlinear optics, which approximates the polarization response by an expansion into a series with respect to the external electric field¹. The guaranteed linearity may be less obvious if a medium undergoes some nonperturbative interaction with a pump pulse (which is not a part of $E(t)$ in the above equations). However, it is well established, and we can refer the referee to some relevant papers:

- “Ultrahigh-Order Wave Mixing in Noncollinear High Harmonic Generation” (<https://doi.org/10.1103/PhysRevLett.106.023001>);
- “First-principles study of ultrafast and nonlinear optical properties of graphite thin films” (<https://doi.org/10.1103/PhysRevB.103.085433>);
- “Dressing effects in the attosecond transient absorption spectra of doubly excited states in helium” (<https://doi.org/10.1103/PhysRevA.91.061403>);

¹ See, e.g., “Effective First-Order Response: The Generalized Linear Susceptibility” in <https://doi.org/10.1146/annurev.pc.43.100192.002433>

- “Simulating Pump–Probe Photoelectron and Absorption Spectroscopy on the Attosecond Timescale with Time-Dependent Density Functional Theory” (<https://doi.org/10.1002/cphc.201201007>).

Silica.

It is undoubted that energy balance requires more photons to be absorbed in order to promote an electron from the valence band to the conduction band in this case. Nevertheless, as the authors write, due to the relatively the strong field used in the experiment, $E = 1.4 \text{ V/\AA}$, the Keldysh parameter is close to 1. Multiphoton absorption is, instead, expected to happen for a Keldysh parameter $\gg 1$ (H. Mashiko et al., Nat. Commun 1468 (2018)).

As we explained above, the model that we used to reconstruct the measured data makes no assumption on whether the physics of photoinjection is closer to the multiphoton or tunneling limit.

The field strength used by the authors is similar to what used by some of them in another work: A. Schiffrin et al., Nature 439, 70-74 (2013), not cited here. Schiffrin showed that important strong-field mechanism, here neglected, influence light-matter interaction and carrier injection in SiO₂. These phenomena do not necessarily follow the n -th power of the field intensity as for the case of multi-photon absorption. The charge due to the field induced polarization (Fig. 2b) seems to scale exponentially with the applied field strength (in the same range as for the present work).

The exact mechanism of photoinjection is irrelevant to all the major claims of our work because our main results are related to the motion of charge carriers after photoinjection ends. The revised manuscript cites A. Schiffrin et al., Nature 439, 70-74 (2013).

In another related work from the authors themselves: M. Schultze et al., Nature 493, 75-78 (2013), not cited in the manuscript, the authors have already shown one of the main results of the current manuscript as they found that for silica: “Both the nonlinear polarization and the conduction band population induced by the strong field return to near-zero immediately after the laser pulse for $F \leq 2.5 \text{ V/\AA}$. This is fully consistent with the abrupt decay of field-induced transient NIR reflectivity and XUV absorption bleaching: see Figs 3b and 2b, respectively. These results imply that the sample exposed to fields as high as $F \leq 2.5 \text{ V/\AA}$ resumes its original (field-free) state immediately after exposure.”

Therefore, it is not too surprising that the Drude-Lorentz model can catch the system response in SiO₂ soon after the interaction with the pulse.

As we explained above, the work that M. Schultze et al. published in 2013 provided no insight into optical-field-driven dynamics of charge carriers after the interaction with the strong field, which is what the present manuscript is about. (We do not regard extreme-ultraviolet light as *optical*.) The fact that excitations caused by a strong pump pulse insignificantly affect the interaction of extreme ultraviolet light with a sample has no implications whatsoever on how an infrared pulse would interact with the photoinjected charge carriers. This consideration refutes the reviewer’s argument.

Finally, in the work from Sommer et al. (Ref. 14 of the present work), the authors show that the nonlinear polarization in silica can have important effects, inducing an energy transfer between the field and the material which lasts for all the pulse duration and it is not confined in the main pulse cycle.

From Fig. 4 in the paper by Sommer *et al.*, we see that the central half-cycle photoinjects most charge carriers. Even with the two half-cycles next to the central one, most charge carriers emerge within an interval as short as 3 femtoseconds. This time interval is shorter than half of the optical oscillation of the test field that we employed in the present work. For such a short photoinjection event, the distortions experienced by the test field are insensitive to the details of the photoinjection process (as we explain in the main text of our manuscript). For this reason, the model that assumes instantaneous photoinjection gives such a good agreement with the measurements in those time intervals where the pump and test pulses do not overlap. Developing that model, we also experimented with models that did not use the assumption of instantaneous photoinjection and learned that these measurements are largely insensitive to how exactly photoinjection occurs.

The theoretical model used to fit the data assumes linear polarization and sudden ionization. Based on what written above, I believe these to approximations to be unjustified.

The reviewer's skepticism is based on two misunderstandings. The first one is the concept of linear response—the response to a weak test pulse is linear even if the response to the strong pump pulse is highly nonlinear. The second one is what exactly justifies the assumption of sudden ionization—it is the confinement of ionization to a half-cycle of the *test* pulse that matters for an appropriate description of the distortions that the test pulse experiences upon propagation through the sample.

While one can state that the non-linear polarization and the finite ionization time window are responsible mainly for the deviation between the model and the data observed in Fig. 4 around 0 fs, I wonder if the agreement at big positive and negative time delays is robust.

At large negative values of $t - \tau$, the test pulse propagates through an unperturbed sample, which means that, in an ideal experiment, there would be no distortions whatsoever. In those time intervals, the measured values of $\Delta E(t)$ are due to noise, fluctuations, drifts, and the uncompressed part of the pump pulse, which makes some weak pump field to be present even tens of femtoseconds before the peak of the pump pulse. The simulated $\Delta E(t)$ is zero as long as t precedes the moment of photoinjection. At large positive values of $t - \tau$, the properties of a photoexcited medium will change because charge carriers will first thermalize and then recombine with each other. These processes take place on a time scale that is much larger than what we show in Fig. 4 of our manuscript. Since our manuscript is focused on what happens during the first 15 femtoseconds after photoinjection, our model does not account for the slow relaxation of charge carriers.

As the authors write in the supplementary information the model has 5 free parameters, which gives quite some room in terms of fitting capability. How robust is the fit? Or, in other words, can the fit reproduce any experimental result?

Our model properly reconstructed all the measured data. Here, it is important to keep in mind that the five parameters were used to fit an entire delay scan. For every particular delay, using our model would result in overfitting, but when the model is applied for all the delays at once, it becomes robust.

The authors write “... we evaluate an upper limit for the formation time of the Drude-Lorentz response as 4 fs—there is no systematic deviation between the measurement and the Drude-Lorentz fit for $t > \tau + 4$ fs”. If we neglect the comments at point 1 and 2

above and try to evaluate the upper limit for the for the formation time of the Drude-Lorentz response, I believe that we should start counting from the time the system displays a non-zero response (interaction with the laser) and not from $t = \tau$ as figure 3f suggests. Here we see that the system response starts roughly 5-6 fs before the black dashed line. Therefore, the total time it takes to reach Drude-Lorentz would be about 10 fs at least.

Waveform distortions that we see 5-6 fs before the middle of the pulse are due to nonlinear wave mixing. At these early times, the pump field is yet too weak to photoinject charge carriers. Therefore, such an early time may not serve as a starting point for the formation of the collective response of photoinjected charge carriers.

The existence of a non-zero signal for some femtoseconds before the black-dashed line in Fig. 3f shows that the carrier injection is not instantaneous in SiO₂. Otherwise, how can the probe waveform be modified if the carriers are yet to be created?

Even before charge carriers are created, a sufficiently strong pump field distorts the field of the test pulse via nonlinear wave mixing, which is described with odd-order nonlinear susceptibilities: $\hat{\chi}^{(3)}$, $\hat{\chi}^{(5)}$ etc. (A hat here denotes a tensor.) Because of this, the nonlinear processes affect the test field even though the pump field is orthogonally polarized. For example, the real part of $\hat{\chi}^{(3)}$ is responsible for cross-phase modulation, affecting the phase of the test pulse. The imaginary part of $\hat{\chi}^{(3)}$ is responsible for attenuating the test field via the process where the simultaneous absorption of one photon from the test pulse and one photon from the pump pulse promotes a valence electron to a conduction band. This particular process is relevant in the case of silicon; it creates charge carriers but does not require a high concentration of them for the nonlinear absorption associated with this wave mixing to have an impact on the test field. That is, the transient absorption due to the imaginary part of $\hat{\chi}^{(3)}$ can be significant where the Drude-Lorentz response is insignificant.

Minor comments:

Since the model cannot reproduce what happens during light-matter interaction and since the carrier injection mechanism is not clear, I find the use of 1-fs-scale potentially misleading. I suggest to remove it.

We have changed the title of our manuscript to “The optical response of solids after 1-fs-scale photoinjection”. Although it still contains the phrase “1-fs-scale”, it makes it clear that our manuscript is about what happens after photoinjection, while we have all reasons to believe that photoinjection was confined to a time interval not longer than a few femtoseconds. The new title also emphasizes that our manuscript is about the properties of solids in the optical domain (rather than extreme ultraviolet).

It is not clear what the 2.8 fs refers to in Fig. 1b. Is it the supposed window in which the carrier injection takes place? How has this been defined?

It is the time between the moment when the green curve reaches 10% of its final value and the moment when it reaches 90% of the final value. We have added the explanation to the caption of Fig. 1.

While the signal in fig. 4 reduces by an order of magnitude when moving from Si to SiO₂, the error bars stay comparable. Is it due to better statistics for the SiO₂ case?

The error bars represent standard deviations evaluated from several delay scans one after another. During the measurement, the field of the test pulse changes due to various drifts

and fluctuations. We diminish their adverse effects by measuring reference waveforms in parallel with the perturbed waveforms and applying our model to these reference waveforms. Small changes in the incident test pulse between delay scans have no effect on the retrieved time-dependent refractive index. However, changes that occur during a delay scan do affect the results, and they are mainly responsible for the error bars in Fig. 4. The error bars in the two sets of measurements are comparable because fluctuations in the test pulse were comparable.

In figure 4 a legend for the open circles and squares is missing.

We have added the missing legends.

The reference style seems not to be uniform. Titles are missing.

We have corrected the reference list.

The work from Sommer on the nonlinear polarization is referenced while talking about instantaneous ionization in the introduction. While I believe the work to be important and to be cited, I do not see the direct link.

Figure 4a in Sommer *et al.* provides evidence that the central half-cycle of an intense 3-fs near-infrared pulse indeed photoinjects most charge carriers. Interpreting that figure, one should keep in mind that the values of the nonlinear work *at zero crossings of the electric field* are most directly related to the creation of charge carriers. In that figure, the positions of zero crossings coincide with the local minima of the nonlinear work. Indeed, in the presence of a strong field, the nonlinear work, $W(t)$, represents not only the energy spent on exciting valence electrons to conduction bands but also the energy spent on creating transient nonlinear polarization, that is, reversibly displacing electrons. Even at the zero crossings, $W(t)$ is only roughly proportional to the concentration of charge carriers because $W(t)$ also accounts for their kinetic energies. With this in mind, Fig. 4a in Sommer *et al.* indeed provides evidence for a single half-cycle photoinjecting most charge carriers.

There is a typo at line 499 when indicating the equation used to evaluate the reconstructed waveforms. Is it possible that “((21)-(23))” should instead be “(20)-(22)”?

We have corrected the mistake; thank you!

Referee #2

The paper reports an sub-cycle-resolved observation of how mid-infrared (MIR) optical field response to the photoexcited solids, silicon and fused silica, during the first few femtoseconds after the instantaneous photo-injection of carriers which is temporally confined to a single dominant half cycle of the electric field. Based on the observation of the modification of the MIR probe electric field from the reference one, the authors concludes that the Drude-Lorentz response forms within a few femtoseconds after the photo-injection, in particular, this time was much shorter than the inverse plasma frequency in the case of fused silica.

In my opinion, there are two important aspects in this manuscript. One is the authors' excellent problem setting to be solved in this work. The fundamental question how fast electron-hole plasma forms in photoexcited solid state materials has been a focus of attention in not only the field of attosecond physics but also solid state physics. This is because the time required electron-hole plasma formation after the initial photoexcitation is closely related to the quantum many-body dynamics in which charge screening play an essential role. Clarifying the dynamics and characteristic time scale

of this plasma formation or charge screening is still challenge to ab initio approach. From not only theoretical but experimental point of view, this is not easy task. Generally, it is predicted to be comparable to the inverse plasma frequency, which corresponds to the time scale of hundreds of attoseconds in metals and tens of femtoseconds in semiconductors. To date, one famous experimental work, which is referred as Ref. 7 in this manuscript, was reported for investigating GaAs semiconductor over 20 years ago. Although the charge screening dynamics has been long-standing mystery, it is still undoubtedly one of the deepest and the most fundamental research targets as an unestablished many-body dynamics in the field solid state physics.

The other important aspect is the authors' advanced spectroscopic technique named as generalized heterodyne optical sampling technique (GHOST). This technique enables us to directly measure the electric field of light in time domain with very simple experimental setup compared to conventional optical field sampling technique such as attosecond streak method. By combining GHOST to pump-probe spectroscopy, the authors achieved the direct measurement of the probe electric field passed through the photo-injected materials. Electric field measurement provides us optical response of real and imaginary part, which correspond to refractive index and absorption coefficient of photo-injected solid state materials

Thus, experimental approach of the electron-hole plasma formation dynamics will be one of the exciting, broad-interest and important targets in the attosecond science. In addition, the carrier screening dynamics will interest many researchers in broad field of solid state physics. Therefore, the scientific importance and impact described in this manuscript is in the scope of Nature.

The referee's interpretation is accurate, demonstrating a good understanding of our work. We are glad that the referee shares our enthusiasm about this piece of research!

However, I find that their main achievements in the current manuscript do not contain enough sufficient interpretation of experimental data for justifying a publication in Nature.

The main conclusion of the authors is derived from the experimental observation that the probe MIR electric field is modified with the photo-injection in Si and SiO₂ sample. My main concern is that it is unclear how the authors distinguish the electron-hole plasma state (collective response under the screened Coulomb collisions) from the free carrier state where unscreened bare Coulomb collisions is dominant in their experimental data.

This is a valid concern, and we give a more detailed explanation below. Very briefly, our results show that the transition between these two states has little effect on how the infrared test field interacts with the photoexcited solid.

In the following I describe my main opinions and concerns in detail.

(1) In my knowledge, the first experimental achievement which clearly showed electron-hole plasma formation in time domain was Ref. 7 (Huber et al., Nature 414, 286 (2001).) They used THz single-cycle electric field as a probe and 100-fs near infrared pulse as a photo-injection pulse, and then investigated electron-hole plasma formation in photoexcited GaAs semiconductor. The experimental concept is very similar with the experimental setup in the current manuscript. In Ref. 7, they measured the dielectric function of the photoexcited GaAs by the THz probe electric field, and concluded the electron-hole plasma formation by confirming the appearance of additional oscillatory signal which originates from the electron-plasma formation. This was supported by the spectral peak of the imaginary part of the inverse dielectric function obtained by the signal matches the estimated plasma frequency. They also confirmed the real part of the

inverse dielectric function well described a dressed Coulomb interaction, which is characteristic of electron-hole plasma formation (Fig. 3 (a) & (b) in Ref. 7).

Here, let us note that the red curves in Fig. 3 (a) & (b) of Ref. 7 represent the Drude fit rather than a sophisticated calculation of a many-body response. The work by Huber *et al.* (Ref. 7) established that the Drude fit starts to accurately describe the measured data as the pump-probe delay approaches 175 fs. They had the possibility of representing it in a clear visual way by plotting the real and imaginary parts of the inverse permittivity, but such a visualization is not necessary for establishing whether the light-driven motion of charge carriers obeys the predictions of the Drude model.

In contrast, the authors did not analyze the modified probe electric field of MIR for obtaining the dielectric function of the photo-injected sample.

For fused silica, the retrieved plasma frequency ($\frac{\omega_{pl}}{2\pi} = 9$ THz) is far outside the spectral range covered by the test pulse, which is from 0.1 PHz to 0.2 PHz.

In analogy to the analysis described in Ref. 7, the modified probe electric field (MIR waveform distortion) must have information of the plasma formation, such as plasma frequency. In Si, the authors describes the plasma period to be 2 – 3 fs, which corresponds to roughly 400 THz. It is unclear why the data shown in the current manuscript does not show any sign for the plasma formation like the data in Fig. 3 in Ref. 7. The differences between the current manuscript and Ref. 7 are (i) the probe wavelength (2.2 micron in this study and THz in Ref. 7), (ii) the sample (silicon and silica in this study and GaAs in Ref. 7), (iii) the intensity of the photo-injection pump pulse (two or five photon excitation in this study and resonance excitation in Ref. 7), and (iv) time scale for measurement (a few femtosecond or less in this study and a few tens of femtosecond to 200 fs in Ref. 7). If you can not distinguish the plasma state from the quasi-free state, it may be difficult to discuss the dynamics of electron-plasma formation and its dynamics. The authors should comment on this point or add a discussion in the revised manuscript.

For silicon, the situation is more nuanced. The retrieved plasma frequency is also outside of the test-pulse spectrum: $\frac{\omega_{pl}}{2\pi} = 0.37$ PHz. However, the frequency where $-\text{Im}[1/\epsilon(\omega)]$ has its main maximum depends on the rate of momentum relaxation, and our measurements indicate a very fast momentum relaxation. Here is a plot of the inverse permittivity for the measurement that appears in the manuscript:

The loss function ($-\text{Im}[1/\epsilon(\omega)]$) has its main maximum at 0.13 PHz, which implies the existence of long-wavelength longitudinal plasma waves at this frequency, and these collective excitations set the times scale for the build-up of screening. We obviously cannot make a plot like that in Huber *et al.* (Ref. 7) because the inverse of this frequency (7.7 fs) is shorter than the duration of the test pulse (the FWHM of intensity is 12 fs). We had to come up with an analysis applicable to the case where the change of optical properties happens within the test pulse, which is what we did.

(2) It seems that the description about the plasma frequency or plasma density is very few, which will support the authors' data interpretation. According to the "Methods" section, the plasma frequency in the Drude and Lorentz model is a fitting parameter. It is likely to be better for readers to get the information of plasma frequency or plasma density which is directly linked to its plasma frequency. In the case of Si, is the plasma frequency on the order of 10^{14} Hz?

For silicon, the retrieved plasma frequency is 0.37 PHz ($\omega_{\text{pl}} = 2.3 \text{ fs}^{-1}$). The Methods section in the revised manuscript contains a table summarizing the retrieved values of the fit parameters (at the end of the subsection titled "Drude-Lorentz model for instantaneous photoinjection").

It is generally much difficult to estimate nonlinear excitation process compared to one-photon resonance excitation of electrons in the conduction band. Estimated plasma density, which can be derived from the plasma frequency, should be comparable to the estimated value from the photoexcited carrier density via nonlinear excitation. The authors should offer the information of plasma frequency to the readers, and discuss about it.

The estimation of the plasma density is somewhat problematic because the average effective mass of charge carriers may significantly change as photoinjection becomes nonperturbative. We studied this dependence theoretically: Qasim *et al.*, Phys. Rev. B **98**, 214304 (2018). Using the same model as in that study, we simulated the interaction of a 3.5-fs 750-nm laser pulse with crystalline silicon. When the pump pulse strongly excites the medium, its peak electric field in the medium can no longer be estimated using Fresnel equations. To overcome this problem, we estimated the field by comparing the simulated plasma frequencies to those that we obtained by analyzing the measured data. From this analysis, we estimate the peak electric field in silicon to be 0.35 V/Å. With this peak field, we estimated that the average effective mass in our silicon measurements was 0.8 atomic units. Combining this value with the retrieved plasma frequency, we estimate the concentration of conduction-band electrons to be $1.3 \times 10^{21} \text{ cm}^{-3}$. This value can be compared with theoretical predictions. In the above-mentioned simulations, it was $1.5 \times 10^{21} \text{ cm}^{-3}$. We also simulated the nonlinear photoinjection using SALMON (<https://salmon-tddft.jp>), which is a code that implements the real-time real-space time-dependent density functional theory. Using the Tran-Blaha meta-GGA exchange with Perdew-Wang correlation, we again obtained an electron concentration of $1.5 \times 10^{21} \text{ cm}^{-3}$.

(3) When you compare the amplitude decrease in Fig. 4 (a) with that in Fig. 4 (b), the scale of the $\text{Im}[n_{\omega}(t)]$ changes from about 0.8 to 1×10^{-3} . This difference must be due to the sample difference. Since $\text{Im}[n_{\omega}(t)]$ is related to the photoexcited carrier density, it is nearly 3-order different between Si and silica. However, the rise time of Si (Fig. 4 (a)) is almost same as that of silica (Fig. 4 (b)), which may be 3 fs. Could you comment on the reason? If highly nonlinear photoexcitation occurs, the rise time of the

photoexcited carrier density must be faster. Is the probe temporal resolution not enough for measuring such rise time?

This is one of the main points in the technical part of our manuscript: a fast change in carrier concentration does not mean an equally fast change in the time-dependent refractive index. To make this point, we had to carefully define what we mean by a time-dependent refractive index (please see the Methods section). Let us finish our reply to this comment with a quote from our manuscript: “We also observe here that neither the theoretical refractive index nor the reconstructed waveforms change abruptly even though photoinjection was modeled as an instantaneous event. This is because, in such measurements, $\Delta E(t)$ is approximately proportional to the electric current induced by the test field, and the current gradually builds up as the field accelerates the charge carriers. For the same reason, the time dependence of $n_\omega(t)$ does not end with photoinjection (see Eq. (16) in the Methods section).”

(4) Fig. 4 (a) shows the onset of the change of the data shown in the upper and lower graphs significantly different. The onset of the amplitude decrease represented by red circle starts about 4 fs earlier than that of the zero-crossing shift represented by blue circle. The slope of the zero-crossing shift is much steeper than that of the amplitude decrease. This tendency is similar in Fig. 4 (b). If the carrier excitation and the phase velocity change are induced by two-photon absorption and optical Kerr effect respectively, the onset and the slope of both curves should be almost identical. The authors can add the discussion about this discrepancy for readers.

Two considerations matter here. The first one is what we wrote replying to the previous comment: the time-dependence of the refractive index differs from that of the charge-carrier concentration. This is also obvious from the theoretical curves in Fig. 4 (a): the solid red curve is steeper than the solid blue curve, although both of them were produced by a model that approximates photoinjection as an instantaneous event at $t = \tau$. The second consideration is about how the Kerr effect and multiphoton absorption manifest themselves when the pump and test pulses overlap. The Kerr effect has an effect opposite to that of charge carriers: it increases the real part of the refractive index, compensating for the decrease caused by electron-hole plasma. In contrast, the imaginary part of the refractive index is increased by both the transient nonlinear absorption of the test pulse (which is a multiphoton process) and the absorption due to interband transitions enabled by the pump pulse. Together these two considerations explain the observation that the reviewer made.

(5) You can recognize a large deviation from the theoretical curve in the delay time from -7 to 0 fs in the lower graph of Fig. 4 (b). It looks very large deviation compared to other differences between experiments and theory shown in Fig. 4. The authors should comment on this point.

The theoretical curves in Fig. 4 are not supposed to match the data at $t < \tau$. Indeed, the Drude-Lorentz model does not describe the nonlinear interaction between the test and pump pulses, while the assumption of instantaneous photoinjection means that there is no Drude-Lorentz response at $t < \tau$. Our motivation for doing so is twofold. First, we want to convey as clearly as possible the following message: even if we assume an instantaneous change in the concentration of charge carriers, the probe-waveform distortions appear gradually, and it takes time for the refractive index to change. Second, the interaction of solids with the pump pulse was in the strong-field regime, where no existing model reliably describes the optical response to the test pulse. (Even those models that the literature refers to as *ab initio* either do not account for relaxation processes or account for them phenomenologically because the strong optical electric field presents a big challenge for more sophisticated modeling of many-

body quantum dynamics.) So we wanted to focus on those outcomes of our measurements that can be analyzed unambiguously.

In summary, my major concern is that I don't see the clear experimental evidence which you can judge the formation of electron-hole plasma, compared to the previous experiment in Ref. 7. Since the novelty and impact of the manuscript depends on the question how quickly the electron-hole plasma formation when the photo-injection confined to sub-cycle time scale, the identifying the plasma formation experimentally is an inevitable problem to be solved for the measurement and the data analysis for satisfying the criteria of Nature publication in my opinion.

We fully agree with the logic of this remark, but we argue that our work provides solid experimental evidence for the formation of the Drude-Lorentz (plasma) response, even though the plasma frequency was outside the bandwidth of the test pulse.

Referee #3

In their work, Zimin and coworkers are investigating the response of bulk materials to ultra-short, femtosecond-scale excitations. They report experimental measurement of the change of the optical properties of silicon and fused silica following this excitation. These changes are analyzed in terms of a Drude-Lorentz response, and it is shown that the this response forms much faster than the inverse plasma frequency.

This work is clearly interesting and timely, and also well written. In particular, it allows for distinguishing the difference between the Drude-Lorentz response and the remaining part of the changes, the latter been qualitatively different for silicon and fused silica.

This is an accurate description of our work, and we are glad about this positive assessment.

However, while the reported results are quite interesting, the referee finds that the paper is sometimes missing some deeper understanding and analysis of the reported results.

For instance, in Fig. 3, silicon and fused silica are behaving qualitatively differently for the non-Drude-Lorentz part of the waveform distortion. It would be good if the authors could give a tentative explanation why this is the case.

The concentration of photoinjected charge carriers was orders of magnitude larger for silicon than for fused silica. Because of this, the nonlinear processes that require the simultaneous presence of the pump and test pulses (e.g., Kerr effect and multiphoton absorption) played a minor role in the silicon case. In the measurements with the fused-silica sample, the pump field was stronger, while the concentration of charge carriers was much smaller. As a consequence, the transient nonlinear response, which forms when both pulses simultaneously interact with the medium, was much more prominent. From Fig. 3 (b), one can deduce whether the transient change of the optical thickness or the transient change in absorption played a major role at $t < \tau$. The time $t = 0$ was chosen as the moment where the average reference waveform had its global maximum, while the pseudocolor diagram shows the difference between the perturbed and unperturbed waveforms. In the area $t < \tau$, there is a blue stripe to the left of the dashed green line and a red stripe to the right of it. This means that photoinjection makes the waveform arrive at a later time; that is, it transiently increases the optical thickness of the sample. This is characteristic of the Kerr effect.

At the moment, it is stated that the observed changes in the central region Fig. 3F is related to the plasma period in fused silica. The reader is therefore left with questions like: Why isn't it also the case for silicon where the plasma period is between 2-3fs

according to the manuscript, and should therefore be detectable? As the samples been of different thicknesses, could this originate from propagation effects?

This may be a misunderstanding—there is no such statement in our manuscript. The term “plasma frequency” is well established, but it may be a bit misleading in the context of the Drude model, which does not predict any plasma oscillations. (In fact, the classical interpretation of the Drude model is based on considering a single free charged particle interacting with an oscillating electric field.) So, no oscillations in Fig. 3 are supposed to be at the plasma frequency. Nevertheless, the inverse plasma frequency is believed to be the characteristic time scale for the build-up of many-body interactions. We merely point out that, for fused silica, we do not see any signatures of this build-up in the optical response of the medium to the test pulse.

Similar lake of understanding is for the Fig. 4 where the authors also note that “We cannot fully explain the magnitude of this shift” but associate this with possible propagation effects. Here, the paper would have really benefited by having measurement performed for different thicknesses, to be able to bring deeper understand of the reported effects.

This critical remark motivated us to model the propagation through the fused-silica sample by numerically solving Maxwell’s equation. Here are the main results of these simulations:

This figure is formatted similarly to Fig. 3 of our manuscript, but it compares propagation through a 100-nm sample of fused silica (left column) to that through a 12.7- μm sample (right column), simulated by solving Maxwell's equations. The results in panels **a** and **b** account for the optical Kerr effect ($\chi^{(3)} = 7 \times 10^{-23} \text{ m}^2 / \text{V}^2$) and the Drude-Lorentz response (using the same parameters as in the manuscript). Panels **c** and **d** were obtained with the Drude-Lorentz response being the sole origin of pump-induced waveform distortions. For the simulation results shown in panels **e** and **f**, we fully suppressed the Drude-Lorentz response. Analyzing the above figure, we realized that we had incorrectly determined the zero delay in the previous version of the manuscript. This mistake is corrected in its revised version. Now the peak increase in the optical thickness of the fused-silica sample almost coincides with the peak of the pump pulse's intensity envelope.

The referee thinks that the results of the Drude-Lorentz should be better analysed and presented. Indeed, it would be interesting to get what are the obtained values after the fit, in particular for the plasma frequencies, and the resulting number of carriers N_D and N_L , to see if the extracted values are meaningful or not. The same is true for the lifetimes γ_D and γ_L .

We have added the retrieved fit parameters to the Methods section. The estimation of charge-carrier concentration can be found in our response to Referee #2. We have considered adding this information to the manuscript and decided against it because of the ambiguity in the average effective mass or charge carriers, which sensitively depends on the pump pulse [M. Qasim *et al.*, Phys. Rev. B 98, 214304 (2018)]. For silicon, we estimated the mass to be $0.8m_e$, but the amorphous nature of fused silica prevents us from making a reliable estimation.

Moreover, why these plasma frequencies are different? Indeed, if one only consider one type of Lorentz oscillators, N_L should correspond to N_D . This is somehow directly implied by the sentence (lines 468-470) "once an electron makes a transition from a valence state to a state in a conduction band, the electron can then be further excited into a higher conduction band by absorbing a photon, while the vacancy left in the valence band can be filled by photoexciting an electron from a deeper valence band".

We allow $\omega_{p1,D}$ to be different from $\omega_{p1,L}$ because describing the absorption of a photoexcited solid with a single Lorentz term is an oversimplification—every single interband transition contributes a Lorentz term [M. Qasim *et al.*, Phys. Rev. B 98, 214304 (2018)]. This oversimplification is acceptable as long as the model properly describes the properties of a photoexcited solid within the spectral range covered by the test pulse, which is the case in our measurements. Nevertheless, $\omega_{p1,L}$, γ_L , and ω_r should be viewed merely as fit parameters that describe a fictitious Lorentz resonance rather than physical quantities related to the concentration of electron-hole pairs, dephasing rate, and a resonance frequency. (In fact, this is true for most published applications of the Drude-Lorentz model.)

As a summary, while the presented results are certainly of interest, the referee finds that the present manuscript does not provide a deep understanding of the reported data, and thus seems to be suited for a publication in Nature.

We do not see how this conclusion follows from the specific criticism raised by the referee. We developed a general theory that describes this kind of experiments (see the Methods section). We could have simply retrieved the time-dependent refractive index using that theoretical framework, but we dug deeper—we identified a simple model that adequately described the optical response to the test pulse after photoinjection. Armed with this model,

we dug even deeper and arrived at the main claims of our work, which the referee did not refute. Any sophisticated experiment has aspects that remain unclear or uncertain, but what matters most is the major claims and whether there are any reasons to doubt them. We admit that we do not fully understand the distortions of test waveforms in the time interval where the pump pulse is present, but our main findings are related to what happens afterward, where we think we deeply understand our results.

Reviewer Reports on the First Revision:

Referees' comments:

Referee #1 (Remarks to the Author):

D. A. Zimin and coworkers present a new version of their manuscript and a point-by-point reply to the Referees' concerns. The reply to Ref. #1 can be summarized in 3 points:

1) Charge injection mechanism and sudden injection assumption. The results reported by the authors in their reply show that a two-photon mechanism is indeed a too crude approximation. The authors state that the exact injection mechanism is not relevant. What matters is the confinement of the charge injection. For this reason, for simplicity, they prefer not to change the content of Fig. 1b. While I agree that the exact injection mechanism may be irrelevant, I believe that the current Fig. 1b is not rigorous and potentially misleading. If a picture is known to be incorrect it should not be used. Especially for a journal as important as Nature. I strongly advise the authors either to produce a "cartoon-like" figure with no quantitative information in it, or to revise the figure by using what they reported in their reply.

2) Non-linear polarization response. I agree with the authors: the polarization induced by the weak test field which is linear. While the authors clearly state this in their reply, it seems that nothing has been done in the text. It would have been better for the authors to add a sentence in the manuscript to make this point clearer and to avoid any possible source of misunderstanding between the effect of the strong field and the effect of the weak one.

3) Robustness of the fitting procedure and the main claims. While the previous two points have been clarified, at least to a certain extent, I am sorry to say that the new version of the manuscript did not manage to clear the doubts on this last point. On contrary, it had the opposite effect. In the new version of the manuscript the panels of figure 3 and 4 referring to SiO₂ have been considerably modified. I think this is probably related to the fact that the authors corrected the choice of their zero as they write in reply to Ref. #2 "Analyzing the above figure, we realized that we had incorrectly determined the zero delay in the previous version of the manuscript. This mistake is corrected in its revised version.". As a consequence of this correction, fig 3f shows a stronger discrepancy between experiment and model for $t < 4$ fs and fig. 4b shows a disagreement between experiment and model for $0 < t < 5$ fs that was not there before. This makes me wonder for two reasons:

- It seems that the graphical representation of the reported results strongly depends on the choice of the time zero. Thus suggesting that the conclusion drawn may not be robust. How accurate is the zero calibration? How this affects the conclusions on a 1-fs time scale?
- Even if the agreement between model and experiment is considerably changed between -5 and +5 fs, it seems that the main finding is not affected: the Drude-Lorentz response is reached within 4 fs. How can it be? The new experimental data in Fig. 4b seems to be shifted by 2.5-3 fs towards the right. Surely a not-negligible value if compared with a 4 fs scale. So how can the conclusions stay unchanged? In other words, how can two different figure 3 and 4 lead to the same conclusions?

I see two possible answers: i) on a fs time scale, the results depend strongly on the choice of the time zero and the conclusions have to be accordingly refined. But this means that the accuracy with which we determine the time zero influences the accuracy and validity of the whole analysis. Or ii), the conclusions do not depend on the exact time dependence of the optical properties, which would suggest that they cannot easily be proved/supported with the current experiment.

In view of this, I am sorry to say that cannot recommend publication.

Referee #2 (Remarks to the Author):

In my first review report, the scientific importance and impact in this manuscript satisfy one of the criteria of Nature publication since the authors' fundamental question how fast electron-hole plasma forms in photoexcited solid state materials has been a focus of attention in not only the field of attosecond physics but also solid state physics. However, I expressed several concerns regarding with the interpretation and the method of experimental data described in the first manuscript, although I found the authors' excellent problem setting to be solved in this work. In the following, I will describe my opinions about the authors' response to my five comments.

(1)The method of how to experimentally distinguish the plasma state from the quasi-free state.

In this manuscript, the main experimental observation are Figs. 2 (a) and (b). This data shows the temporal shift and the amplitude decrease of the test MIR electric field when the instantaneous carrier photoinjection from valence to conduction band of the sample by using the mono-cycle NIR pump pulse. Although the shift and decrease compared to the reference MIR electric field, which corresponds to that without the photoinjection, was clearly detected, the essential point is what the shift and the amplitude decrease originates from.

The method the authors proposed is to extract the distortion of the observed waveform from the waveform component which can be explained by the Drude-Lorentz model (corresponding to the electron-hole plasma state). The authors obtained this Drude-Lorentz-model-based component by fitting the measured data in the temporal window of $t > \tau + 5$ fs to the theoretically derived time-dependent refractive index based on the Drude-Lorentz model. The "tau" is the time of the instantaneous photoinjection by the mono-cycle pump NIR pulse to the center of the test MIR electric field. Therefore, the authors' method is essentially based on the assumption that the MIR waveform 5fs after the photoinjection corresponds to the component which can be explained by the Drude-Lorentz model. This indicates that the initial waveform distortion within 5 fs after photoinjection is equivalent the component which CAN NOT be explained by the Drude-Lorentz model, which corresponds to the quasi-free state. If the authors measure the same situation by 100-fs MIR test pulse, and assume that Drude-Lorentz-model-based component corresponds to the temporal window $t > \tau + 50$ fs, there is a possibility to change the conclusion.

This is my main concern about how to experimentally distinguish the plasma state from the quasi-free state. In the previous pioneering work by the Ref. 7 (Huber et al., Nature 414, 286 (2001).), they

confirmed the plasma state formation by measuring the spectral peak of the imaginary part of the inverse dielectric function, which roughly matches the estimated plasma frequency. They also confirmed the real part of the inverse dielectric function well described a dressed Coulomb interaction, which is characteristic of electron-hole plasma formation (Fig. 3 (a) & (b) in Ref. 7).

In the authors' response, they shows a plot of the inverse permittivity for the measurement that appears in this manuscript. At $\tau = -20$ fs and -40 fs, the loss function ($-\text{Im}[1/\epsilon]$) shows a clear peak at 0.13 PHz. What happens on the peak at $\tau = -10, -5, 0, +5$ fs and so on? If this peak clearly decreases around $\tau = 0$ fs compared to $\tau = -20$ and -40 fs, this signal seems to be clear evidence for the plasma formation. Can the authors estimate the plasma formation time from the build-up time of the peak experimentally? Since the duration of the test MIR test pulse is 12 fs, the authors should be able to measure the temporal evolution of the peak within the first 12 fs.

(2) The description about the plasma frequency or plasma density

The authors' correction satisfy my comment. It is beneficial to readers to show the table summarizing the information about the retrieved plasma frequency.

(3) and (4) The rise time of the photoexcited carrier density in Figs. 4 (a) and (b)

I understand the authors' answer that a fast change in carrier concentration does not mean an equally fast change in the time dependent refractive index. However, is there any possibility that the response time of the time-dependent refractive index is depend on the cycle period of the MIR test pulse? Is the MIR test pulse insufficient to resolve such instantaneous charge injection? This indicates that the temporal resolution of this experimental setup limits to the one-cycle period of the MIR test pulse.

(5) The large deviation from theoretical curve of the data in Fig. 4 (b)

The authors explain my misunderstanding about this question.

In summary, the response and the revision is partially enough for resolving my concerns regarding the authors' data analysis described in my concerns of (2) to (5). However, my main concern (1) is not fully resolved, and the response from the authors does not seem to directly answer this main concern.

As described in the comment in (1), one of the clear evidence of the Drude-Lorentz-response formation seems to be the appearance of the peak at 0.13 PHz. My suggestion is that the authors show the temporal dependence of the build-up of the peak. The other suggestion is that they show the pump NIR peak intensity dependence. If the authors show the pump NIR peak intensity dependence of the Drude-Lorentz-response-formation interval, in particular, for silicon sample, it will more strongly support the validity of their method than the current version of the manuscript. If the plasma frequency is reduced by decreasing the pump NIR peak intensity, the waveform

deviation around the center of the test MIR pulse should clearly appear than the current Fig. 4 (e).

Referee #3 (Remarks to the Author):

In their reply, the authors have addressed all my points. They replied to my opinion satisfactorily and made clearer some of the points that I did not properly understand before.

They also added new data and information that improved the quality of the manuscript.

I therefore now recommend the manuscript for publication.

Author Rebuttals to First Revision:

Response to reviewer's comments

Referee #1

D. A. Zimin and coworkers present a new version of their manuscript and a point-by-point reply to the Referees' concerns. The reply to Ref. #1 can be summarized in 3 points:

1) Charge injection mechanism and sudden injection assumption. The results reported by the authors in their reply show that a two-photon mechanism is indeed a too crude approximation. The authors state that the exact injection mechanism is not relevant. What matters is the confinement of the charge injection. For this reason, for simplicity, they prefer not to change the content of Fig. 1b. While I agree that the exact injection mechanism may be irrelevant, I believe that the current Fig. 1b is not rigorous and potentially misleading. If a picture is known to be incorrect it should not be used. Especially for a journal as important as Nature. I strongly advise the authors either to produce a "cartoon-like" figure with no quantitative information in it, or to revise the figure by using what they reported in their reply.

In the revised manuscript, Fig. 1b shows the nonlinear work obtained using the time-dependent density functional theory. The details of these calculations are provided in a new section that we have added to Methods: "Nonlinear work".

2) Non-linear polarization response. I agree with the authors: the polarization induced by the weak test field which is linear. While the authors clearly state this in their reply, it seems that nothing has been done in the text. It would have been better for the authors to add a sentence in the manuscript to make this point clearer and to avoid any possible source of misunderstanding between the effect of the strong field and the effect of the weak one.

We have changed "As long as the polarization response to the test pulse is linear" to "While the interaction with the pump pulse is nonlinear, the polarization response to the weak test pulse is linear." If there were no word limit, we would gladly elaborate more on this topic in the main text. However, our main challenge in revising the manuscript has been to shorten it toward the 2700-word limit. For this reason, a detailed discussion of the concept of linearity is only possible within the Methods section of the paper (where it is already included).

3) Robustness of the fitting procedure and the main claims. While the previous two points have been clarified, at least to a certain extent, I am sorry to say that the new version of the manuscript did not manage to clear the doubts on this last point. On contrary, it had the opposite effect. In the new version of the manuscript the panels of figure 3 and 4 referring to SiO₂ have been considerably modified. I think this is probably related to the fact that the authors corrected the choice of their zero as they write in reply to Ref. #2 "Analyzing the above figure, we realized that we had incorrectly determined the zero delay in the previous version of the manuscript. This mistake is corrected in its revised version.". As a consequence of this correction, fig 3f shows a stronger discrepancy between experiment and model for $t < 4$ fs and fig. 4b shows a disagreement between experiment and model for $0 < t < 5$ fs that was not there before. This makes me wonder for two reasons:

- It seems that the graphical representation of the reported results strongly depends on the choice of the time zero. Thus suggesting that the conclusion drawn may not be robust. How accurate is the zero calibration? How this affects the conclusions on a 1-fs time scale?*

We have added a new section to Methods: “Timing of the injection event”. In this section, we explain how delay zero was determined. This method allows for a $\lesssim 1$ -fs accuracy, but it does not exclude the possibility that the delay may drift by several femtoseconds during a measurement that takes several hours. Preparing the very first version of our manuscript, we had to choose what to believe: the zero-delay calibration in the raw data, which we thought meant an unphysical delay in the moment where the Drude-Lorentz response began to form, or our understanding of the physics. Since we could not exclude a systematic error, we chose the latter. Working on the previous resubmission, we clarified the role of propagation effects and found that the zero delay had been correctly determined during the measurements. Our current manuscript combines our best efforts to understand the physics while utilizing all available experimental data, including those not used in the first analysis.

• Even if the agreement between model and experiment is considerably changed between -5 and +5 fs, it seems that the main finding is not affected: the Drude-Lorentz response is reached within 4 fs. How can it be? The new experimental data in Fig. 4b seems to be shifted by 2.5-3 fs towards the right. Surely a not-negligible value if compared with a 4 fs scale. So how can the conclusions stay unchanged? In other words, how can two different figure 3 and 4 lead to the same conclusions?

This is a good question. The answer consists of two parts, underlying to which is our focus on the charge-carrier motion after the central part of the pump pulse. First, within just a few femtoseconds, relaxation makes the electric current that is induced by the test pulse independent of the exact moment of photoinjection (as one can see from Fig. 2b). So, for delays that exceed the maximum relaxation time ($\max\{\gamma_D^{-1}, \gamma_L^{-1}\}$), waveform distortions are insensitive to small changes in the moment of photoinjection. The second part of the answer is about the fact that our estimation of how long it takes for the Drude-Lorentz response to form in fused silica has not changed once we corrected the zero delay. It is simply because, preparing the very first version of the manuscript, we wanted to give a safe upper limit for this time. If you look at Fig. 4b in that version, you will see that it would have been reasonable to state that the solid lines and the error bars coincide already starting from $t - \tau = 1$ fs. However, we saw a regular pattern below the thick black dashed line in Fig. 3f and put the grey dots where that pattern ended. Comparing Fig. 4b in the initial submission to that in the other two submissions, you can see that the main difference between them consists in the error bars being shifted to the right. (We also re-optimized the parameters of the Drude-Lorentz model, but this had a minor effect on the solid curves.) Now there are obviously big differences between the error bars and the solid curves for $t - \tau \lesssim 4$ fs, which are also consistent with what we see in Fig. 3f. It is just a coincidence that we did not have to change our estimation of the formation time even by a femtosecond.

I see two possible answers: i) on a fs time scale, the results depend strongly on the choice of the time zero and the conclusions have to be accordingly refined. But this means that the accuracy with which we determine the time zero influences the accuracy and validity of the whole analysis. Or ii), the conclusions do not depend on the exact time dependence of the optical properties, which would suggest that they cannot easily be proved/supported with the current experiment.

What matters here is that the waveform distortions during the central part of the pump pulse depend on processes that are outside the scope of the present manuscript: the optical Kerr effect, the dynamical Franz-Keldysh effect, and all the wave-mixing processes that make a strong pump field influence the ability of a test field to accelerate charge carriers. Future pump-probe field-resolved measurements promise insights into these phenomena, and

reliable calibration of the zero delay will be of paramount importance for these future measurements. It is less important for the present manuscript, which demonstrates the new technique and where the focus is on the motion of charge carriers after photoinjection, in the time interval where the pump pulse is too weak to influence this motion.

In view of this, I am sorry to say that cannot recommend publication.

We hope that our arguments convince you that our analysis is indeed robust and our conclusions are indeed reliable.

Referee #2

In my first review report, the scientific importance and impact in this manuscript satisfy one of the criteria of Nature publication since the authors' fundamental question how fast electron-hole plasma forms in photoexcited solid state materials has been a focus of attention in not only the field of attosecond physics but also solid state physics. However, I expressed several concerns regarding with the interpretation and the method of experimental data described in the first manuscript, although I found the authors' excellent problem setting to be solved in this work. In the following, I will describe my opinions about the authors' response to my five comments.

(1)The method of how to experimentally distinguish the plasma state from the quasi-free state. In this manuscript, the main experimental observation are Figs. 2 (a) and (b). This data shows the temporal shift and the amplitude decrease of the test MIR electric field when the instantaneous carrier photoinjection from valence to conduction band of the sample by using the mono-cycle NIR pump pulse. Although the shift and decrease compared to the reference MIR electric field, which corresponds to that without the photoinjection, was clearly detected, the essential point is what the shift and the amplitude decrease originates from.

The method the authors proposed is to extract the distortion of the observed waveform from the waveform component which can be explained by the Drude-Lorentz model (corresponding to the electron-hole plasma state). The authors obtained this Drude-Lorentz-model-based component by fitting the measured data in the temporal window of $t > \tau + 5$ fs to the theoretically derived time-dependent refractive index based on the Drude-Lorentz model. The "tau" is the time of the instantaneous photoinjection by the mono-cycle pump NIR pulse to the center of the test MIR electric field. Therefore, the authors' method is essentially based on the assumption that the MIR waveform 5fs after the photoinjection corresponds to the component which can be explained by the Drude-Lorentz model. This indicates that the initial waveform distortion within 5 fs after photoinjection is equivalent the component which CAN NOT be explained by the Drude-Lorentz model, which corresponds to the quasi-free state. If the authors measure the same situation by 100-fs MIR test pulse, and assume that Drude-Lorentz-model-based component corresponds to the temporal window $t > \tau + 50$ fs, there is a possibility to change the conclusion.

This is my main concern about how to experimentally distinguish the plasma state from the quasi-free state. In the previous pioneering work by the Ref. 7 (Huber et al., Nature 414, 286 (2001).), they confirmed the plasma state formation by measuring the spectral peak of the imaginary part of the inverse dielectric function, which roughly matches the estimated plasma frequency. They also confirmed the real part of the inverse dielectric function well described a dressed Coulomb interaction, which is characteristic of electron-hole plasma formation (Fig. 3 (a) & (b) in Ref. 7).

In the authors' response, they shows a plot of the inverse permittivity for the measurement that appears in this manuscript. At $\tau = -20$ fs and -40 fs, the loss function ($-\text{Im}[1/\epsilon]$) shows a clear peak at 0.13 PHz. What happens on the peak at

tau = -10, -5, 0, +5 fs and so on? If this peak clearly decreases around tau = 0 fs compared to tau = -20 and -40 fs, this signal seems to be clear evidence for the plasma formation. Can the authors estimate the plasma formation time from the build-up time of the peak experimentally? Since the duration of the test MIR test pulse is 12 fs, the authors should be able to measure the temporal evolution of the peak within the first 12 fs.

Here are the plots of $\text{Re}[1/\varepsilon]$ and $-\text{Im}[1/\varepsilon]$ for several delays in the measurements on silicon, where the permittivity ε was evaluated from waveform distortions as if the medium were static (ignoring the fact that the pump pulse quickly changes the permittivity, which then keeps evolving for a time that is longer than the duration of our pump pulse).

Specifically, we obtained the above results by numerically solving

$$\frac{E_{\omega}^{\text{out}}}{E_{\omega}^{\text{in}}} = \frac{1}{\cos\left(\frac{\omega}{c}\sqrt{\varepsilon_{\omega}}d\right) - \frac{i}{2}\left(\sqrt{\varepsilon_{\omega}} + \frac{1}{\sqrt{\varepsilon_{\omega}}}\right)\sin\left(\frac{\omega}{c}\sqrt{\varepsilon_{\omega}}d\right)},$$

which is equivalent to Eq. (11) in Methods. As long as the pump and test pulses overlap, the dependence of this quantity on delay is mainly an artifact of applying the analysis designed for static media to a medium with rapidly changing optical properties. In particular, the pump pulse must arrive at least 20 femtoseconds after the peak of the test pulse for ε to approach the unperturbed permittivity. Similarly, if the pump pulse arrives first, we cannot regard ε as the true permittivity if the delay is smaller than ~ 20 fs. To verify this conclusion, we apply the very same analysis to the waveforms that we obtained with the Drude-Lorentz model, where the properties of the medium change abruptly at $t = 0$ and do not change after that. The results are similar to those we see in the measured data:

Drude-Lorentz model with instantaneous photoinjection

This is why we cannot draw any conclusions about the formation time of the plasma state by applying the analysis that is valid only when a medium does not (significantly) change its properties during the probe pulse.

We would also like to point out that there is no evidence in the pioneering results by Huber *et al.* [Nature 414, 286 (2001)] that the optical response of the “quasi-free” state cannot be approximated by a Drude (or Drude-Lorentz) model. In that paper, the authors show that $1/\epsilon$ evolves until a certain moment (inverse plasma frequency), at which the Drude model provides an accurate fit. From the published results, we infer that the Drude model would probably provide an accurate fit also at smaller delays as long as the pump and probe pulses do not overlap. The transition from the quasi-free state to the plasma state then makes the parameters of the Drude model evolve in time until they reach their steady values. An attempt to fit the entire delay scan (for non-overlapping pulses) with a single set of parameters would provide a good fit quality only for a limited range of delays.

(2) The description about the plasma frequency or plasma density The authors' correction satisfy my comment. It is beneficial to readers to show the table summarizing the information about the retrieved plasma frequency.

This was indeed a good comment.

(3) and (4) The rise time of the photoexcited carrier density in Figs. 4 (a) and (b)

I understand the authors' answer that a fast change in carrier concentration does not mean an equally fast change in the time dependent refractive index. However, is there any possibility that the response time of the time-dependent refractive index is depend on the cycle period of the MIR test pulse? Is the MIR test pulse insufficient to resolve such instantaneous charge injection? This indicates that the temporal resolution of this experimental setup limits to the one-cycle period of the MIR test pulse.

Yes, indeed: although the temporal resolution of these measurements can be much better than the duration of the test pulse, it cannot be much better than the time it takes a photoinjection electron to reach its maximal velocity along the test field. This time is a half of the test field's optical cycle (4 fs in our case). Judging from the data in Fig. 4, our temporal resolution is probably slightly better than that, but it is not sufficient to resolve the dynamics of photoinjection, especially in the measurements with the fused-silica sample. For this purpose, a test pulse with a higher frequency of optical oscillations would be necessary.

(5) The large deviation from theoretical curve of the data in Fig. 4 (b)

The authors explain my misunderstanding about this question.

In summary, the response and the revision is partially enough for resolving my concerns regarding the authors' data analysis described in my concerns of (2) to (5). However, my main concern (1) is not fully resolved, and the response from the authors does not seem to directly answer this main concern. As described in the comment in (1), one of the clear evidence of the Drude-Lorentz-response formation seems to be the appearance of the peak at 0.13 PHz. My suggestion is that the authors show the temporal dependence of the build-up of the peak. The other suggestion is that they show the pump NIR peak intensity dependence.

The temporal dependence is in the plot that we inserted in response to one of the previous comments. Here are the plots of $1/\epsilon$ for several values of the peak pump field in the case of the silicon sample:

Note that this is a separate set of measurements, which is not used in our manuscript. In the delay scan that we chose for the manuscript, the peak field of the incident pump pulse was 0.79 V/\AA . In the measurements where we gradually increased $E_{\text{pump}}^{\text{peak}}$ performing one delay scan after another, we see how the peak in the loss function ($-\text{Im}[1/\epsilon]$) gradually shifts toward higher frequencies, just as one would expect. We do not have a definite explanation for why that peak in the 0.79-V/\AA measurement (the plots on page 4) was observed at a frequency that is below that of the peak in the 0.64-V/\AA measurement. These measurements were performed on different days. Most likely, this is due to subtle, day-to-day differences in the alignment of the experimental setup.

If the authors show the pump NIR peak intensity dependence of the Drude-Lorentz-response-formation interval, in particular, for silicon sample, it will more strongly support the validity of their method than the current version of the manuscript. If the plasma frequency is reduced by decreasing the pump NIR peak intensity, the waveform deviation around the center of the test MIR pulse should clearly appear than the current Fig. 4 (e).

Although we do have several measurements where $-\text{Im}[1/\varepsilon]$ has a peak within the bandwidth of the test pulse, we insist that this analysis is meaningful only if there is no significant overlap between the pump and test pulses.

Referee #3

In their reply, the authors have addressed all my points. They replied to my opinion satisfactorily and made clearer some of the points that I did not properly understand before.

They also added new data and information that improved the quality of the manuscript.

I therefore now recommend the manuscript for publication.

We are glad that we properly addressed the points raised by Referee #3.

Reviewer Reports on the Second Revision:

Referees' comments:

Referee #1 (Remarks to the Author):

I recognize that the authors have improved the manuscript since its first version, especially with regards to the actual carrier injection. Nevertheless, I cannot recommend its publication. While reading their reply I found myself sincerely confused by them stating: *"...In this section, we explain how delay zero was determined. This method allows for a \lesssim 1-fs accuracy, but it does not exclude the possibility that the delay may drift by several femtoseconds during a measurement that takes several hours. Preparing the very first version of our manuscript, we had to choose what to believe: the zero-delay calibration in the raw data, which we thought meant an unphysical delay in the moment where the Drude-Lorentz response began to form, or our understanding of the physics."*

I do not think that in a well-conceived scientific experiment one should be forced to "choose what to believe". If the setup does not allow excluding **a several fs drift during a measurement**, an honest data representation should reflect this. Which means that the experimental points in figure 4 should have horizontal error bars reporting the experimental uncertainty. Those should be of several fs, seriously challenging the idea that the setup has the time resolution needed to follow any dynamics on a few-fs scale.

Referee #2 (Remarks to the Author):

In my second review report, I raised four concerns. In the third response letter and the revised version, the authors made an effort to give me sincere and clear answers to my concerns that I could not understand in previous versions. Especially, they gave a detailed explanation about my concern of (1). They showed that the real and imaginary permittivity which was evaluated from their measured waveform distortions shown in Fig. 2. According to these figures, I understood the reason why the authors cannot draw any conclusions from the analysis.

In summary, since I cannot find any points for not justifying a publication in Nature, the current manuscript seems to satisfy the criteria of Nature.